# Enhanced gradient-based MCMC in discrete spaces

**Benjamin Rhodes**  *ben.rhodes@ed.ac.uk*
*University of Edinburgh*

**Michael U. Gutmann**  *michael.gutmann@ed.ac.uk*
*University of Edinburgh*

**Reviewed on OpenReview:** *https://openreview.net/forum?id=j2Mid5hFUJ*

## Abstract

The recent introduction of gradient-based Markov chain Monte Carlo (MCMC) for discrete spaces holds great promise, and comes with the tantalising possibility of new discrete counterparts to celebrated continuous methods such as the Metropolis-adjusted Langevin algorithm (MALA). Towards this goal, we introduce several discrete Metropolis-Hastings samplers that are conceptually inspired by MALA, and demonstrate their strong empirical performance across a range of challenging sampling problems in Bayesian inference and energy-based modelling. Methodologically, we identify why discrete analogues to *preconditioned* MALA are generally intractable, motivating us to introduce a new kind of preconditioning based on auxiliary variables and the 'Gaussian integral trick'.

## 1 Introduction

Gradient-based Markov Chain Monte Carlo (MCMC) offers an efficient, robust way to sample from a wide-class of probability distributions. The gradient serves as a concise descriptor of local geometry, which can be exploited when designing MCMC transition operators. In continuous spaces, such operators are often based on the Langevin diffusion (Roberts & Tweedie, 1996; Roberts & Rosenthal, 1998) or Hamiltonian Monte Carlo (Duane et al., 1987; Neal et al., 2011), which can be unified under a single complete framework (Ma et al., 2015). Until recently, gradient-based operators were only viable for continuous distributions—after all, the standard gradient is undefined

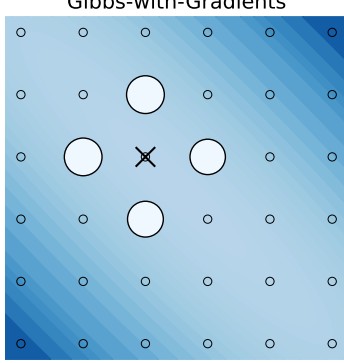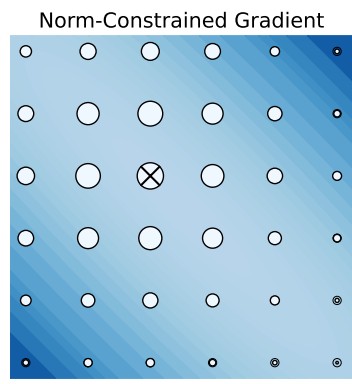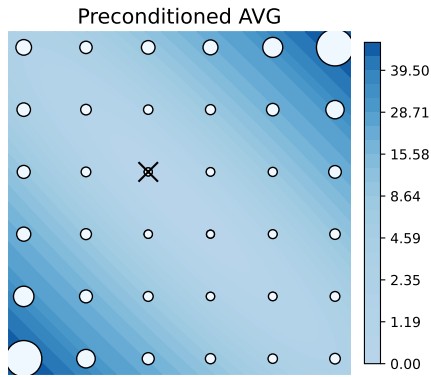

Figure 1: Metropolis-Hastings proposal distributions for an existing gradient-based sampler (left) and our proposed ones (centre & right). The blue contours delineate a continuous function that, when restricted to the discrete lattice, equals the target log-probability function (up to a constant). The white circles represent the proposal distribution given the current state of the Markov chain (black X). Larger circles mean higher probability mass. The new samplers can update multiple dimensions at a time, which can lead to more efficient mixing of the Markov chains. In addition, preconditioned AVG (right) accounts for the strong positive correlation between the two dimensions.

for discrete domains. However, Grathwohl et al. (2021) made the important observation that many probability mass functions are naturally thought of as the *restriction* of a continuous function (defined in e.g. $\mathbb{R}^d$) to a discrete subset (e.g. $\mathbb{N}^d$). Gradients in this ambient space can thus inform the design of a transition operator in the restricted discrete space. They demonstrate this via a promising new method, Gibbs-with-Gradients (GWG), that can be seen as a gradient-based version of the 'locally informed proposals' introduced by Zanella (2020).

Whilst the use of gradient-information makes GWG appealing, the method has multiple unresolved limitations. It:

(L.1) cannot update more than one dimension at a time.
(L.2) does not exploit second-order interactions in the target distribution to define the proposal distribution.

In this paper, we will address these shortcomings via the introduction of several new discrete gradient-based samplers. Our strategy will be to define discrete analogues to *continuous* gradient-based samplers that do not have these limitations. Specifically, we consider the celebrated Metropolis-Adjusted Langevin Algorithm (MALA), which is free from limitation L.1, and its preconditioned variant (PMALA), which is free from L.2. 'Discretising' these samplers is not a straightforward task: (P)MALA is typically viewed through the lens of stochastic differential equations (SDEs), which are not easily translated into discrete state-spaces. Fortunately, we identify two non-SDE characterisations of (P)MALA that are more readily imported into the discrete setting, resulting in the following methods (illustrated in Figure 1):

- The **Norm-Constrained Gradient (NCG)** sampler is constructed by viewing MALA as a locally-informed proposal (Zanella, 2020) whose domain is restricted to a discrete space.[1] NCG replaces the Hamming-ball constraint in Gibbs-with-Gradients with a soft norm constraint, enabling multiple dimensions to be updated at a time, thereby addressing limitation L.1. Unfortunately, preconditioning this sampler in a manner analogous to preconditioned MALA does not appear possible, leaving limitation L.2 unresolved.

- The **Auxiliary-Variable Gradient (AVG)** sampler is constructed by viewing MALA as a marginalised auxiliary sampler (Titsias & Papaspiliopoulos, 2018) whose domain is restricted to a discrete space. The resulting sampler is conceptually and algebraically similar to NCG, but incurs extra computations that make it less appealing. However, AVG has the major advantage that it can be extended to include a kind of preconditioning, thereby addressing limitation L.2. Intriguingly, this **Preconditioned AVG (PAVG)** generalises an existing auxiliary-variable scheme (Martens & Sutskever, 2010) that is only applicable to pairwise Markov Random fields (a.k.a. Boltzmann machines).

The value of these new methods is two-fold: 1) Empirically, they show superior sampling efficiency compared to several important baselines across a range of problems. In particular, they outperform Gibbs-with-Gradients, validating our hypothesis that L.1 and L.2 are indeed limitations that should be addressed. 2) Methodologically, they demonstrate how new discrete gradient-based samplers can be derived via the frameworks of Zanella (2020) and Titsias & Papaspiliopoulos (2018). To our knowledge, we are the first paper to use the latter framework for discrete problems, and our derivation of PAVG shows that auxiliary variables open up fundamentally new possibilities for constructing gradient-based samplers.

## 2 Background

Throughout this work, we will be dealing with distributions $p(\mathbf{s})$ that are only available up to proportionality i.e.

$$\log p(\mathbf{s}) = f(\mathbf{s}) - \log Z \tag{1}$$

where $f$ is a known function defined on a set $\mathcal{S}^d$ and the value of $Z$ is presumed unknown. Depending on the context these distributions may be continuous, with $\mathcal{S} = \mathbb{R}$ and $Z = \int_{\mathbb{R}^d} \exp(f(\mathbf{s}))d\mathbf{s}$, or discrete, with $\mathcal{S} \subset \mathbb{R}$ and $Z = \sum_{\mathbf{s} \in \mathcal{S}^d} \exp(f(\mathbf{s}))$. In the discrete setting, our exposition focuses on two important cases: binary vectors for which $\mathcal{S} = \{0, 1\}$ and finite ordinal data for which $\mathcal{S} = \{s^1, \dots, s^k\}$ is an ordered set of real numbers. However, in Appendix A we show that our methods also easily extend to one-hot vectorised categorical data.

---

[1] The NCG method was independently discovered by Zhang et al. (2022) (first made public on Arxiv on the 20th June 2022); see related work discussion in Section 4.

For continuous distributions, it is common to assume $f$ is differentiable with derivative $\nabla f$. In discrete settings, the standard definition of a derivative is inapplicable and the notation $\nabla f$ is undefined. However, the recent work of Grathwohl et al. (2021) provide a useful definition of $\nabla f$ in discrete settings that applies to many (but not all) functions $f$. Their key observation is that many functions that accept discrete inputs also accept continuous inputs: for instance an Ising model has the form $f(\mathbf{s}) = \boldsymbol{b}^T \mathbf{s} + \mathbf{s}^T J \mathbf{s}$, where the elements of $\mathbf{s}$ are usually constrained to $\{-1, +1\}$, but if we instead assumed them to be real-valued, $f$ would still be a valid function. So many discrete functions can be *uniquely* extended to $\mathbb{R}^d$ by leaving their mathematical form intact, and only altering the domain. If we further assume that this unique extension is differentiable with derivative $\nabla f$, then we can use this as the definition of the 'gradient' of discrete-input function $f$.

## 2.1 Metropolis-Hastings

A common approach to sampling such distributions is to use Metropolis-Hastings (MH) Metropolis et al. (1953); Hastings (1970), which evolves a chain of samples by iteratively sampling a proposal $\mathbf{s}_{t+1} \sim q(\mathbf{s}_{t+1}|\mathbf{s}_t)$ and accepting it with probability

$$\min\left\{1, \frac{\exp(f(\mathbf{s}_{t+1}))}{\exp(f(\mathbf{s}_t))} \frac{q(\mathbf{s}_t \mid \mathbf{s}_{t+1})}{q(\mathbf{s}_{t+1} \mid \mathbf{s}_t)}\right\}. \tag{2}$$

The key challenge here is the choice of the proposal distribution $q$. Roughly speaking, there are four desiderata (Robert et al., 1999) i) tractable to sample ii) tractable to evaluate iii) yields reasonable acceptance rates and iv) the resulting chains have weak dependencies between successive states. Satisfying all four criteria is difficult, especially in high dimensions. Most 'obvious' approaches fail to meet at least one criterion; for instance a uniform distribution over neighbouring states of $s_t$ can satisfy the first three criteria, but fail dramatically on the last.

## 2.2 MALA and its preconditioned variant

Metropolis-Adjusted Langevin Algorithm (MALA) (Roberts & Tweedie, 1996; Dwivedi et al., 2018) is an effective *continuous-space* MH sampler that uses a gradient-based proposal distribution of the form

$$q_\epsilon(\mathbf{s} \mid \mathbf{s}_t) = \mathcal{N}\left(\mathbf{s}; \mathbf{s}_t + \frac{\epsilon}{2}\nabla f(\mathbf{s}_t), \epsilon\boldsymbol{I}\right). \tag{3}$$

where $\epsilon$ is a tunable step-size. If we were to skip the MH accept/reject step, then iteratively sampling this proposal distribution corresponds to a discrete-time simulation of an SDE whose stationary distribution is the target $p(\mathbf{s})$. Such time-discretisations induce errors which the MH step corrects for.

Many target distributions exhibit strong second-order interactions between dimensions. A natural way to handle this is via a suitable linear change of coordinates; a technique known as *preconditioning*. Preconditioned MALA (PMALA) (Roberts & Stramer, 2002) has a proposal distribution of the form

$$q_\epsilon(\mathbf{s} \mid \mathbf{s}_t) = \mathcal{N}\left(\mathbf{s}; \mathbf{s}_t + \frac{\epsilon}{2}M\nabla f(\mathbf{s}_t), \epsilon M\right), \tag{4}$$

where $M$ is a user-specified symmetric positive-definite matrix. This proposal also corresponds to a discretised SDE whose stationary distribution is $p(\mathbf{s})$.

## 2.3 Locally-informed proposals

Zanella (2020) proposed a different framework for incorporating local geometry into an MH proposal distribution, with the advantage of being applicable to discrete spaces. They define *pointwise-informed proposals* of the form

$$q(\mathbf{s} \mid \mathbf{s}_t) = \frac{g(\exp(f(\mathbf{s}) - f(\mathbf{s}_t)))K_\sigma(\mathbf{s} \mid \mathbf{s}_t)}{Z_g(\mathbf{s}_t)} \tag{5}$$

where $g$ is a non-negative 'balancing' function, $K_\sigma$ is a symmetric kernel whose 'width' is controlled by $\sigma$ (e.g. a uniform distribution over a local ball of radius $\sigma$) and $Z_g(\mathbf{s}_t)$ is a normalising constant.

The intuition here is that we take an 'uninformed' kernel and re-weight it according to the target distribution, biasing our proposal in favour of higher density regions. Zanella (2020) identify a class of balancing functions $g$ that are optimal when the kernel is sufficiently local (i.e. $\sigma$ is small); an important member of that class is the square-root function $g(x) = \sqrt{x}$.

### 2.4 Gibbs-with-Gradients: GWG

A major challenge when using locally-informed schemes in discrete spaces is the cost of normalising and sampling the proposal distribution. Even for the most simple choice of kernel—a uniform distribution over a Hamming ball of radius 1—the cost is $\mathcal{O}(d)$ evaluations of $f$ for a $d$-dimensional problem. To reduce this cost, Grathwohl et al. (2021) use the innovative trick of treating $f$ as a continuous function and approximating it with a first-order Taylor expansion about the current state $\mathbf{s}_t$, giving a *first-order informed proposal* of the form

$$q(\mathbf{s} \mid \mathbf{s}_t) = \frac{\exp\left(\frac{1}{2}\nabla f(\mathbf{s}_t)^T(\mathbf{s} - \mathbf{s}_t)\right)\mathbb{I}(\mathbf{s} \in H_1(\mathbf{s}_t))}{Z(\mathbf{s}_t)} \tag{6}$$

where $Z(\mathbf{s}_t) := \sum_{\mathbf{s} \in H_1(\mathbf{s}_t)} \exp\left(\frac{1}{2}\nabla f(\mathbf{s}_t)^T(\mathbf{s} - \mathbf{s}_t)\right)$ and $H_1(\mathbf{s}_t)$ is a Hamming ball of radius 1 around $\mathbf{s}_t$ (i.e. those points that differ from $\mathbf{s}_t$ in a single dimension). This proposal distribution is generally inexpensive to normalise and sample as the dominant cost comes from a single gradient computation $\nabla f(\mathbf{s}_t)$. Once we have computed this gradient, the normaliser is just a sum over $d$ inexpensive terms. These $d$ terms are then reused for sampling: we simply sample from a categorical distribution with $d$ states (representing which dimension of $\mathbf{s}_t$ we will change) with probabilities given by the $d$ possible values of $\exp\left(\frac{1}{2}\nabla f(\mathbf{s}_t)^T(\mathbf{s} - \mathbf{s}_t)\right)/Z(\mathbf{s}_t)$ for $\mathbf{s} \in H_1(\mathbf{s}_t)$.

There are two significant limitations of this proposal distribution: (L.1) For each gradient computation, we can only update *one* dimension. Updating one dimension at a time can be slow, and a straightforward approach to 'broaden' the GWG proposal by using a larger Hamming ball is prohibitively expensive—see Appendix C for more details (L.2) Neither the linear approximation of the target distribution nor the Hamming ball kernel account for second-order interactions between dimensions. That is, equation 6 contains no cross-terms $\mathbf{s}_i\mathbf{s}_j$.

## 3 Discrete analogues to MALA

Our goal in this section is to design new discrete gradient-based MCMC schemes that overcome the aforementioned limitations. We achieve this by 'importing' two different characterisations of the Metropolis-adjusted Langevin Algorithm (MALA) into a discrete setting.

### 3.1 Norm-constrained gradient sampler: NCG

As discussed, the use of Hamming balls in Gibbs-with-Gradients makes it challenging to update more than one dimension at a time. Instead, we propose to use a soft norm 'constraint', transforming equation 6 into

$$q_\epsilon(\mathbf{s} \mid \mathbf{s}_t) \propto \exp\left(\frac{1}{2}\nabla f(\mathbf{s}_t)^T(\mathbf{s} - \mathbf{s}_t)\right)\exp\left(-\frac{1}{2\epsilon}\|\mathbf{s} - \mathbf{s}_t\|_2^2\right). \tag{7}$$

Such a first-order informed proposal was already discussed by Zanella (2020) in the context of *continuous spaces*, where, after normalising, it corresponds exactly to the proposal distribution used by Metropolis-adjusted Langevin Algorithm (MALA) in equation 3. Our key insight is that the functional form of this proposal is still valid in discrete spaces. Moreover, an important benefit of this functional form is that it can be *factorised* across all dimensions. After factorising and rearranging[2] terms, we obtain[3]

$$q_\epsilon(\mathbf{s} \mid \mathbf{s}_t) = \prod_{i=1}^{d} \sigma\left(\left[\frac{1}{2}\nabla f(\mathbf{s}_t)_i + \frac{1}{\epsilon}s_{t,i}\right]s_i - \frac{1}{2\epsilon}s_i^2\right), \qquad \sigma(x) := \frac{\exp(x)}{\sum_{\mathcal{S}}\exp(x)} \tag{8}$$

Due its factorised structure, this proposal distribution is efficient to evaluate and sample, making it straightforward to use in a Metropolis-Hastings scheme; see Algorithm B.1 for pseudocode and Appendix C for further discussion of

---

[2] When rearranging, we can ignore any terms in the exponential that do not depend on $\mathbf{s}$, since they are absorbed into the normaliser.
[3] The definition of $\sigma(x)$ abuses notation by assuming x is a function of $s \in \mathcal{S}$, and that we sum over all values of s in the denominator.

computational cost. Like MALA, it has a step-size parameter $\epsilon$ that controls the 'width' of the distribution. Larger step-sizes enable proposals $\mathbf{s}_{t+1}$ that differ from $\mathbf{s}_t$ in multiple dimensions; this is a key property that GWG lacks. Finally, we note that NCG should be viewed as an alternative to (and not a generalisation of) GWG, since there is no setting of the step-size $\epsilon$ that recovers GWG.

### 3.1.1 The intractability of preconditioning NCG

It's natural to wonder: if discretising MALA yields NCG, what does discretising preconditioned MALA yield? Unfortunately, we find that this question leads to a *dead end*. To see why, we first note that PMALA, just like MALA, can be viewed as a first-order informed proposal by replacing the Euclidean norm $\|\mathbf{s} - \mathbf{s}_t\|^2$ in equation 7 with a Mahalanobis norm, giving

$$q_\epsilon(\mathbf{s} \mid \mathbf{s}_t) \propto \exp\left(\frac{1}{2}\nabla f(\mathbf{s}_t)^T(\mathbf{s} - \mathbf{s}_t)\right) \exp\left(-\frac{1}{2\epsilon}(\mathbf{s} - \mathbf{s}_t)^T M^{-1}(\mathbf{s} - \mathbf{s}_t)\right). \tag{9}$$

Specifically, we obtain PMALA by noting that, for $\mathbf{s} \in \mathbb{R}^d$, the right-hand side can be rearranged and normalised to obtain the Gaussian distribution in equation 4. However, when we restrict $\mathbf{s}$ to a discrete domain, this proposal distribution is a pairwise Markov random field that, in general, is intractable to normalise and sample from, making it unusable within a Metropolis-Hastings scheme. The key point here is that log-quadratic distributions are tractable in $\mathbb{R}^d$, but this tractability vanishes as soon as we alter the domain.

One could avoid this intractability by choosing a highly restricted, sparse form for $M^{-1}$ (e.g. diagonal[4]). We prefer to instead focus on the more general problem of constructing a gradient-based sampler that can work with arbitrary $M$. To solve this challenge, we depart from the locally-informed framework of Zanella (2020) and introduce an alternative MH-framework that operates in an extended state-space. This framework also contains a discrete analogue to MALA that resembles NCG, with the key distinction that it admits a kind of tractable preconditioning.

### 3.2 Auxiliary Variable Gradient sampler: AVG

We follow Titsias & Papaspiliopoulos (2018) in showing how MALA can be derived as part of a *continuous*-space auxiliary variable framework. Subsequently, we show how to translate the method into a discrete space.

First, our continuous state $\mathbf{s} \in \mathbb{R}^d$ is augmented with Gaussian auxiliary variables $\mathbf{z} \in \mathbb{R}^d$, to give an unnormalised target density $\pi(\mathbf{s}, \mathbf{z}) = \exp(f(\mathbf{s})) \, \mathcal{N}(\mathbf{z}; \mathbf{s}/\sqrt{\epsilon/2}, \mathbf{I})$. In theory, this distribution could be sampled in a block-Gibbs fashion via alternate sampling of $\mathbf{z}_t \sim \mathcal{N}(\mathbf{z}; \mathbf{s}_t/\sqrt{\epsilon/2}, \mathbf{I})$, and $\mathbf{s}_{t+1} \sim \pi(\mathbf{s} \mid \mathbf{z}_t) \propto \pi(\mathbf{s}, \mathbf{z}_t)$. However, for general functions $f$ this second sampling step is intractable, so it is replaced with an MH accept-reject step using the proposal distribution:

$$q_\epsilon(\mathbf{s} \mid \mathbf{z}_t, \mathbf{s}_t) \propto \exp(f(\mathbf{s}_t) + \nabla f(\mathbf{s}_t)^T(\mathbf{s} - \mathbf{s}_t)) \, \mathcal{N}(\mathbf{z}_t; \mathbf{s}/\sqrt{\epsilon/2}, \mathbf{I}) \tag{10}$$

where we have approximated $\pi(\mathbf{s}, \mathbf{z}_t)$ via a Taylor expansion of $f(\mathbf{s})$. After rearranging and normalising, we obtain

$$q_\epsilon(\mathbf{s} \mid \mathbf{z}_t, \mathbf{s}_t) = \mathcal{N}(\mathbf{s}; \sqrt{\epsilon/2}\mathbf{z}_t + (\epsilon/2)\nabla f(\mathbf{s}_t), (\epsilon/2)\mathbf{I}), \tag{11}$$

Equipped with this proposal distribution, block-wise MH sampling can then be performed in the extended state space. Alternatively, the latent variables could be marginalised out, which yields the MALA proposal in equation 3

$$q_\epsilon(\mathbf{s} \mid \mathbf{s}_t) = \int \mathcal{N}(\mathbf{z}; \mathbf{s}_t/\sqrt{\epsilon/2}, \mathbf{I})q_\epsilon(\mathbf{s} \mid \mathbf{z}_t, \mathbf{s}_t)d\mathbf{z} = \mathcal{N}(\mathbf{s}; \mathbf{s}_t + \frac{\epsilon}{2}\nabla f(\mathbf{s}_t), \epsilon\mathbf{I}). \tag{12}$$

We propose to apply the auxiliary variable procedure just described to discrete state-spaces. Our starting point is equation 10, which is still well-defined if we assume $\mathbf{s}$ belongs to $\mathcal{S}^d \subset \mathbb{R}^d$ and the gradient operator is defined as in Section 2 i.e. using the discrete Taylor approximation 'trick' of Grathwohl et al. (2021). However, if we now rearrange and normalise equation 10, we no longer obtain a factorised Gaussian distribution, but a different kind of factorised distribution $\prod_{i=1}^d q_{\epsilon,i}(\mathrm{s}_i \mid \mathrm{z}_{t,i}, \mathbf{s}_t)$ where each factor has the form

$$q_{\epsilon,i}(\mathrm{s}_i \mid \mathrm{z}_{t,i}, \mathbf{s}_t) = \sigma\left(\left[\nabla f(\mathbf{s}_t)_i + \sqrt{\frac{2}{\epsilon}}\mathrm{z}_{t,i}\right]s_i - \frac{1}{\epsilon}s_i^2\right), \qquad \sigma(\mathrm{x}) = \frac{\exp(\mathrm{x})}{\sum_{\mathcal{S}}\exp(\mathrm{x})}. \tag{13}$$

---

[4]This possibility was explored by the concurrent work of Zhang et al. (2022).

We can easily evaluate and sample this proposal distribution, enabling us to perform block-wise MH sampling as summarised in Algorithm B.2. We refer to this method as the Auxiliary Variable Gradient (AVG) sampler. In appendix D we provide some intuition on how to interpret the auxiliary variables used in AVG via a toy example.

Instead of using block-wise MH, one might try to marginalise out $\mathbf{z}_t$, giving $q_\epsilon(\mathbf{s}\,|\,\mathbf{s}_t) = \prod_{i=1}^d \mathbb{E}_{\mathbf{z}_{t,i}}\left[q_{\epsilon,i}(\mathbf{s}_i\,|\,z_{t,i},\mathbf{s}_t)\right]$. However, we do not use such a marginalised proposal distribution in this paper, since the required expectations have no closed-form solution, necessitating careful numerical approximations. More fundamentally, we focus on the block-wise scheme as it admits an effective kind of preconditioning.

### 3.3   Preconditioned Auxiliary Variable Gradient sampler: PAVG

If the auxiliary-variable view of MALA yields AVG, what does this view imply for preconditioned MALA? Unfortunately, much like in Section 3.1.1, this route of enquiry encounters difficulties. As we explain in Appendix E, we can obtain PMALA by changing the choice of conditional Gaussian in our derivation of MALA above, but doing so sabotages the tractability of the corresponding discrete proposal distribution.

Fortunately, we now show that a different kind of 'preconditioning' is viable. We start by noting that there were essentially two ingredients in our derivation of AVG: i) a local approximation of $f(\mathbf{s})$ and ii) a choice of conditional auxiliary distribution. We will alter both of these choices in the subsequent derivation. First, we make a second-order approximation

$$f(\mathbf{s}) \approx f(\mathbf{s}_t) + \nabla f(\mathbf{s}_t)^T(\mathbf{s} - \mathbf{s}_t) + (1/2)(\mathbf{s} - \mathbf{s}_t)^T M(\mathbf{s} - \mathbf{s}_t), \tag{14}$$

where the second-order term uses a *global* (independent of $t$) symmetric positive definite matrix $M$. The nature of this approximation, and how to choose $M$, is discussed below in Section 3.3.1.

We then define an unnormalised joint distribution $\pi(\mathbf{s}, \mathbf{z}) = \exp(f(\mathbf{s}))\,\mathcal{N}(\mathbf{z}; M_\epsilon^{1/2}\mathbf{s}, \mathbf{I})$, where $M_\epsilon := M + (2/\epsilon)\mathbf{I}$. Just like in our derivation of AVG, block-Gibbs sampling of $\mathbf{z}$ and $\mathbf{s}$ is prevented by the intractable $\pi(\mathbf{s}|\mathbf{z}_t) \propto \pi(\mathbf{s}, \mathbf{z}_t)$ and so we replace this intractable sampling step with an MH accept/reject step. Leveraging equation 14, we approximate $\pi(\mathbf{s}\,|\,\mathbf{z}_t)$ with the following proposal distribution

$$q_\epsilon(\mathbf{s}\,|\,\mathbf{z}_t, \mathbf{s}_t) \propto \exp(f(\mathbf{s}_t) + \nabla f(\mathbf{s}_t)^T(\mathbf{s} - \mathbf{s}_t) + (1/2)(\mathbf{s} - \mathbf{s}_t)^T M(\mathbf{s} - \mathbf{s}_t))\,\mathcal{N}(\mathbf{z}; M_\epsilon^{1/2}\mathbf{s}, \mathbf{I}) \tag{15}$$

$$\propto \exp\left(\left[\nabla f(\mathbf{s}_t) - M\mathbf{s}_t + M_\epsilon^{1/2}\mathbf{z}_t\right]^T \mathbf{s} - (1/\epsilon)\mathbf{s}^T\mathbf{s}\right). \tag{16}$$

$$= \prod_{i=1}^d \sigma\left(\left[\nabla f(\mathbf{s}_t)_i - (M\mathbf{s}_t)_i + (M_\epsilon^{1/2}\mathbf{z}_t)_i\right]\mathbf{s}_i - \frac{1}{\epsilon}\mathbf{s}_i^2\right), \qquad \text{where} \quad \sigma(\mathbf{x}) = \frac{\exp(\mathbf{x})}{\sum_{\mathcal{S}}\exp(\mathbf{x})}, \tag{17}$$

which is fully-factorised and thus tractable to evaluate and sample. The resulting block-wise MH sampling scheme is called Preconditioned AVG (PAVG) and is summarised in Algorithm B.3. By comparing equation 13 and equation 17, we see that setting $M = 0$ recovers AVG.

The key step in obtaining a factorised proposal distribution occurs when going from equation 15 to equation 16, since the cross-terms ($\mathbf{s}_i\mathbf{s}_j, i \neq j$) cancel out. Using Gaussian auxiliary variables to induce such cancellations has a long history in statistical physics where it is known as the Hubbard-Stratonovich transform (Hubbard, 1959) or the "Gaussian integral trick" (Hertz et al., 1991). The trick has also been used in the Machine learning literature; first by Martens & Sutskever (2010) and then extended by Zhang et al. (2012). In both these prior works however, the target function was exactly quadratic i.e. $f(\mathbf{s}) = \mathbf{b}^T\mathbf{s} + \mathbf{s}^T M\mathbf{s}$, in which case the proposal distribution above is 'exact' and an MH accept/reject step is unnecessary. Thus, the proposed PAVG approach can be seen as an extension of Martens & Sutskever (2010) to more general target distributions.[5]

### 3.3.1   Choice of preconditioning matrix $M$

The above derivation assumed an unusual kind of quadratic approximation of $f(\mathbf{s})$ in equation 14 that mixed a local first-order Taylor approximation with a global second-order term. A more standard approximation would use

---

[5]Important caveat: Martens & Sutskever (2010) allow $M$ to have negative eigenvalues. We can also allow this, with small modifications to our proposal distributions as described in Appendix F.

a second-order Taylor approximation. Our foremost reason for avoiding this option is that Hessian computations typically cost $\mathcal{O}(d)$ evaluations of $f$, which would greatly increase of the cost of one iteration of PAVG, which, with a fixed matrix M, only costs $\mathcal{O}(1)$ evaluations of $f$.

Intuitively, one may hope that the approximation in equation 14 is reasonable whenever there are global second-order interactions in the target distribution. We propose to capture these second-order interactions by building on ideas from adaptive continuous-space MCMC (Rosenthal et al., 2011), where one can, for instance, estimate an empirical covariance or precision matrix $M_{emp}$ from a 'dataset' of samples $\mathcal{D}$ obtained during a burn-in period. Given this matrix, we then use a common procedure (e.g. see Algorithm 4 of Andrieu & Thoms (2008)) of re-scaling it by an adaptively learned parameter $\gamma$. We give full details in Appendix F. A benefit of this re-scaling is that we can automatically 'fallback' to AVG by learning $\gamma = 0$. However, in practice, we find that values of $\gamma$ far from 0 are learned. As a final remark, we would like to emphasise that we have no strong apriori reason for thinking that scaled covariance/precision matrices should satisfy equation 14; nevertheless, our empirical results show that PAVG works robustly across a range of scenarios and always outperforms the special case of AVG. Future work may well benefit from finding more theoretically principled choices of M that (unlike Hessians) are cheap to compute.

## 4 Related Work

Most closely related to our work is the concurrent paper of Zhang et al. (2022), who introduce a discrete 'Langevin-like' sampler called DMALA that coincides exactly with NCG. Their empirical results agree with ours in showing that NCG/DMALA is an effective sampler that robustly outperforms GWG. Beyond this shared method, our two works diverge: Zhang et al. (2022) explore variants of DMALA that drop the Metropolis accept-reject step, or use *diagonal* preconditioning; in contrast, we explore a new MALA-inspired auxiliary variable framework that admits a generic *non-diagonal* type of preconditioning (PAVG). In addition, our work is the first to identify the intractability of general preconditioning matrices within the DMALA/NCG framework, as explained in Section 3.1.1.

A range of prior works have 'mapped' discrete spaces to continuous ones, thereby enabling MALA/HMC in the new continuous space. These methods are generally specialised to particular forms of the target distribution (Zhang et al., 2012), or to certain data types, e.g. binary data (Pakman & Paninski, 2013) or trees (Dinh et al., 2017). More problematically, the embedded distributions are typically piece-wise continuous (Nishimura et al., 2017) and highly multi-modal, which may explain their limited empirical success (Zanella, 2020; Grathwohl et al., 2021). One interesting recent advance that may address the aforementioned problems is given by Jaini et al. (2021) who use SurVAE flows to transform a complex embedded distribution into an approximately Gaussian one. The drawback of such an approach is the requirement of training a potentially complex normalising flow model prior to running an MCMC sampler.

## 5 Experiments

We evaluate the newly proposed methods—NCG, AVG & PAVG— on four problem types: 1) sampling from highly correlated ordinal mixture distributions 2) a sparse Bayesian variable selection problem 3) estimation of Ising models and 4) sampling a deep energy-based model parameterised by a convolutional neural network.

Our key baselines are Gibbs-with-Gradients (GWG) Grathwohl et al. (2021) and a standard Gibbs sampler (Geman & Geman, 1984). For our ordinal experiments, we also compare to a Metropolis-Hastings sampler with a uniform proposal over a local ball of radius $r$ and a simple effective extension of GWG for ordinal data that we introduce: ordinal-GWG. In Appendices G & H, we provide details on these baselines and our methodology for tuning step-size parameters to maximise $\|\mathbf{s}_{t+1} - \mathbf{s}_t\|_1$ between successive MCMC states, similar to e.g. Levy et al. (2018).

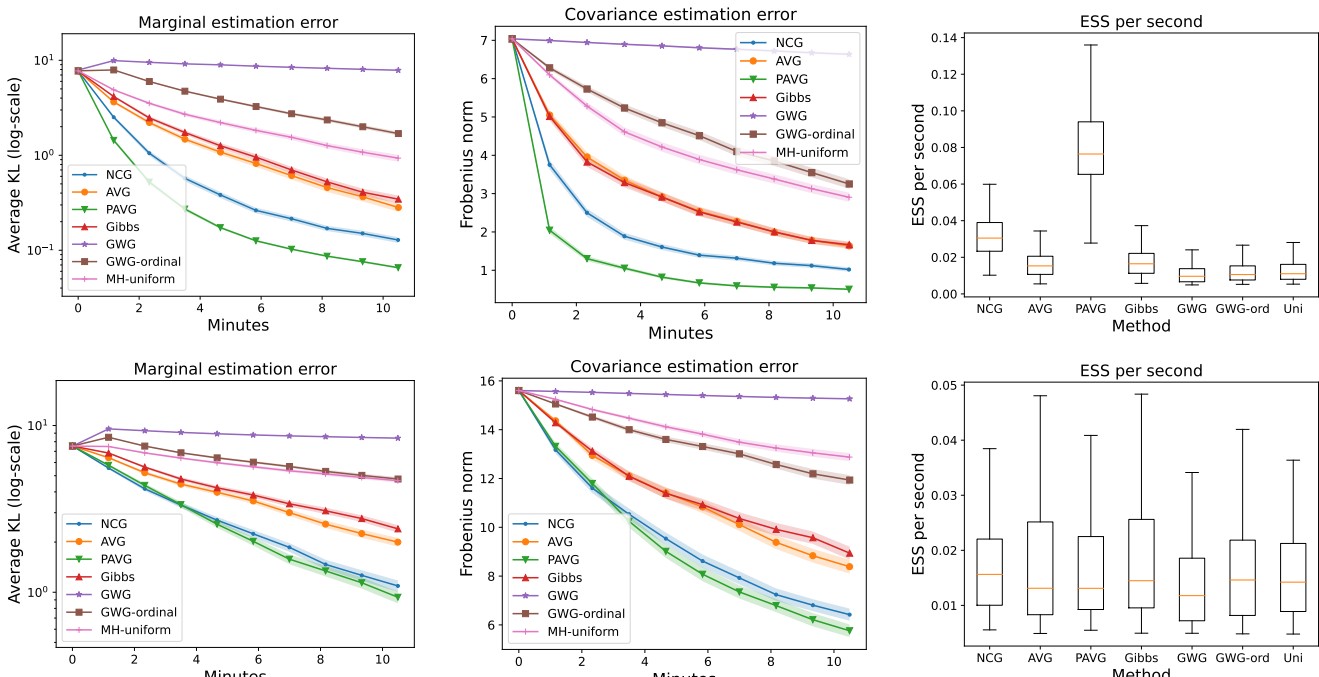

Figure 2: 20D mixture-of-polynomial results. Top row: results for $2^{\text{nd}}$ order polynomial in equation 19. Bottom row: results for $4^{\text{th}}$ order polynomial in equation 20. Left: KL divergence between true and estimated marginals, averaged over all dimensions. Middle: Estimation error of the empirical covariance matrix (lower is better). Right: Effective sample size per second (higher is better). All error bars computed across 100 parallel chains.

### 5.1 20D ordinal mixture-of-polynomials

We define highly-correlated ordinal target distributions $\log p(\mathbf{s}) = f(\mathbf{s}) - \log Z$ over 20-dimensional lattices $\mathcal{S}^{20} \subset \mathbb{R}^{20}$, where $\mathcal{S}$ contains 50 equally spaced points in the interval $[-1.5, 3.0]$. We construct these target distributions from mixtures of fully-factorised distributions, enabling exact sampling and evaluation of the normaliser $Z$. $f$ equals

$$f(\mathbf{s}) = \log\Big(\sum_{k=1}^{50}\exp\Big(\sum_{i=1}^{20}g_k(\mathbf{s}_i)\Big)\Big) \qquad (18)$$

where $k$ indexes a component of the mixture distribution, and $g_k : \mathcal{S} \to \mathbb{R}$ is a polynomial that we allow to take one of two forms:

$$(\mathbf{2^{nd}\ order}) \qquad g_k(\mathbf{u}) = 1.5 - 2\mathbf{t}_k - 6\mathbf{t}_k^2, \qquad \mathbf{t}_k := \mathbf{u} + k/25 \qquad (19)$$

$$(\mathbf{4^{th}\ order}) \qquad g_k(\mathbf{u}) = -\mathbf{t}_k + \mathbf{t}_k^2 - \mathbf{t}_k^3 - \mathbf{t}_k^4, \qquad \mathbf{t}_k := (2\mathbf{u} - 1 + 3k/50). \qquad (20)$$

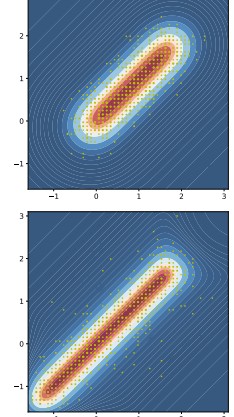

Figure 3: 2D ordinal illustrations; equation 19 & equation 20 correspond to top & bottom, respectively.

We visualise 2D versions of the resulting target distributions in Figure 3. The two dimensions are highly correlated with probability mass concentrating along the main diagonal. Similarly, in 20 dimensions, mass concentrates along the diagonal of a hypercube, meaning *all* dimensions are positively correlated.

We track the similarity between the empirical distribution of MCMC samples $q(\mathbf{s})$ and the target distribution $p(\mathbf{s})$ using the following two metrics i) marginal estimation error: the average KL-divergence between marginals $(1/d)\sum_{i=1}^{d} D_{KL}(q_i \parallel p_i)$ and ii) covariance estimation error: the difference, in Frobenius norm, between the empirical covariance matrices estimated under both distributions. We also track the Effective Sample Size (ESS) of each sampler. Full details of these evaluation metrics are provided in Appendix J.

### 5.1.1 Results

Figure 2 shows the results, with the top & bottom rows corresponding to the $2^{\text{nd}}$ & $4^{\text{th}}$ order polynomials, respectively. In the $2^{\text{nd}}$ order case, the evaluation metrics imply the following ranking:

$$\text{PAVG} > \text{NCG} > \text{AVG} = \text{Gibbs} > \text{MH-uniform} > \text{GWG-ordinal} > \text{GWG} \tag{21}$$

This ranking shows that all newly proposed samplers are either competitive or superior to baselines, with the preconditioned sampler, PAVG, showing especially strong performance. It is slightly surprising that GWG-based methods would perform *worse* than a standard Gibbs sampler, however this is partly due to higher wall-clock costs; GWG-ordinal actually matches Gibbs *per-iteration*; see Appendix I.

For the $4^{\text{th}}$ order polynomial (bottom row), the estimation error metrics (columns 1 & 2) imply a similar ranking as before. The main difference is that PAVG and NCG are now matched in performance. This relative reduction in the performance of PAVG is not entirely surprising: the derivation of PAVG relied on a kind of global quadratic approximation of $f(\mathbf{s})$ (equation 14). Intuitively, it makes sense that our approximation degraded with the addition of $3^{\text{rd}}$ and $4^{\text{th}}$ order terms in the polynomial of equation 20. The ESS metric in the third column provides a less specialised indicator of performance that doesn't use knowledge of the ground truth target distribution. Nevertheless, we see that it provides a similar ranking of methods in the top row, and ranks all methods as more-or-less equal in the bottom row.

## 5.2 100D Sparse Bayesian linear regression

A common use-case of MCMC is sampling posterior distributions that arise in Bayesian analysis. We consider a Bayesian treatment of sparse variable selection in linear regression models, using an experimental setup inspired by that of Titsias & Yau (2017). Given an $n \times d$ design matrix $X$, the response $\boldsymbol{y} \in \mathbb{R}^n$ is modelled as

$$\boldsymbol{y} = X(\mathbf{s} \odot \boldsymbol{\omega}) + \sigma\boldsymbol{\nu} \qquad\qquad \boldsymbol{\nu} \sim \mathcal{N}(0, \boldsymbol{I}_n), \tag{22}$$

where $\boldsymbol{\omega} \in \mathbb{R}^d$ is a vector of weights and $\mathbf{s} \in \mathcal{S}^d$ is a binary random vector that masks out certain covariates. We place a conjugate normal-inverse-gamma prior over weights and noise-variance $(\boldsymbol{\omega}, \sigma^2)$, which can then be analytically marginalised out. Combined with a sparsity-promoting prior over $\mathbf{s}$, we obtain an unnormalised expression for the posterior over binary masks $p(\mathbf{s} \mid X, \boldsymbol{y})$—see Appendix K for a complete description. This posterior tells us which covariates are 'relevant' for predicting the response $\boldsymbol{y}$, and which are 'irrelevant'. Thus, it can be viewed as a kind of sparse variable selection procedure.

This posterior is a 20D distribution over binary vectors, implying $\sim$ 1m possible states, which we can sum over (enabling normalisation). To make the sampling problem more challenging, whilst keeping normalisation tractable, we append additional 'irrelevant' dimensions by multiplying the posterior with $\prod_{i=1}^{80} \mathcal{B}(0.001)$, where $\mathcal{B}$ is the Bernoulli distribution. The fact that we can normalise (and sample) the resulting 100D target distribution, $p(\mathbf{s})$, enables us to compare it to the empirical distribution of MCMC samples $q(\mathbf{s})$ using the following

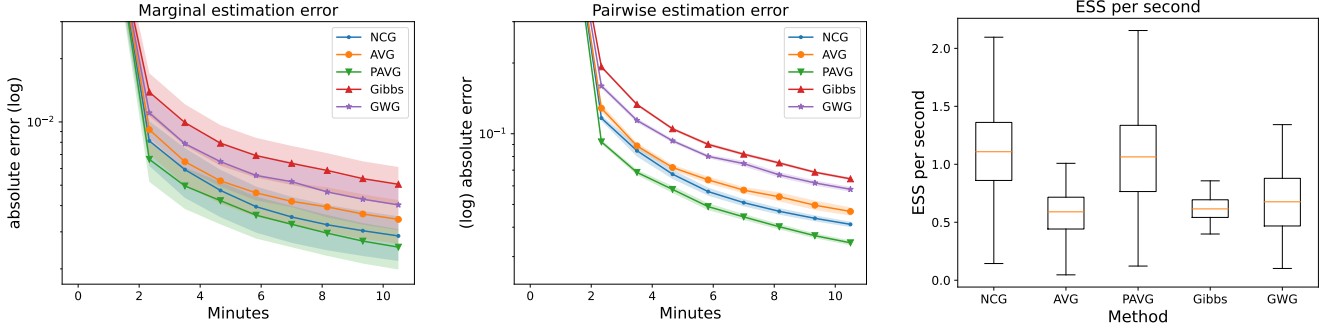

Figure 4: 100D Sparse Bayesian linear regression results. The proposed samplers (NCG, AVG & PAVG) converge faster as measured by bivariate marginals (middle) and are competitive or better as measured by univariate marginals (left) and Effective Sample Size (ESS) (right). Smaller errors and larger ESS indicate better performance.

Table 1: Estimation error of an Ising Lattice matrix learned with persistent contrastive divergence, with standard deviations across 5 runs in parentheses. Each run uses a different seed for all sources of randomness.

| | K=1* | K=5 | K=10 | K=15 | K=20 |
|---|---|---|---|---|---|
| NCG | 0.837 (±.05) | **0.117** (±.0003) | **0.117** (±.0004) | **0.117** (±.0005) | **0.117** (±.0005) |
| AVG | 5.535 (±.002) | 5.452 (±.003) | 5.273 (±.007) | 4.814 (±.029) | 0.120 (±.0006) |
| PAVG model-specific† | **0.120** (±.003) | **0.118** (±.0003) | **0.118** (±.0003) | **0.118** (±.0004) | **0.118** (±.0005) |
| PAVG model-agnostic | 5.521 (±.002) | 0.124 (±0.0005) | 0.121 (±0.0004) | **0.119** (±0.0004) | **0.119** (±0.0007) |
| GWG | 5.271 (±.0038) | 0.163 (±.0009) | 0.138 (±.0004) | 0.132 (±.0002) | 0.128 (±.0008) |
| Gibbs | 4.825 (±.002) | 0.805 (±.003) | 0.167 (±.0005) | 0.136 (±.0003) | 0.132 (±.0005) |

\* K refers to the number of MCMC steps used by *AVG*; suitable multipliers ensure each method has the same budget.

† This corresponds precisely to the Block-Gibbs sampler introduced by Martens & Sutskever (2010).

two metrics i) marginal estimation error: $(1/d)\sum_i^d |q_i(s_i = 1) - p_i(s_i = 1)|$ and ii) pairwise estimation error: $\frac{1}{d^2}\sum_{i,j}^d \sum_{k,l\in\{0,1\}} |q_{i,j}(s_i = k, s_j = l) - p_{i,j}(s_i = k, s_j = l)|$. These metrics are described more fully in Appendix J.

### 5.2.1 Results

Figure 4 shows the results. Under all evaluation metrics, the methods rank as follows: PAVG $\geq$ NCG $\geq$ AVG $\geq$ GWG $\geq$ Gibbs, with the inequalities being strict for the pairwise-error metric. This ranking matches our findings from the previous experiment, except for the swapping of GWG and Gibbs. Most interestingly, we continue to see a clear benefit from preconditioning, even though the target log-probability is far from quadratic.

### 5.3 Estimation of Ising models via persistent-contrastive divergence

Pairwise undirected graphical models are an important class of distributions used in physics, proteomics and economics (MacKay, 2003; Lapedes et al., 1999; Sornette, 2014). Here, we consider the Ising model

$$\log p(\mathbf{s}) = \mathbf{b}^T\mathbf{s} + \frac{1}{2}\mathbf{s}^T J\mathbf{s} - \log Z, \qquad\qquad \mathbf{s} \in \{-1, 1\}^d \qquad (23)$$

where $J$ is a binary matrix multiplied by a constant and $Z$ is the (generally intractable) normaliser.

Following Grathwohl et al. (2021), we define a 100-dimensional Ising model with ground-truth parameters $\mathbf{b}^* = 0$ and $J^*$ is set to a cyclic lattice as depicted in Appendix M. We then obtain 'ground-truth' samples from this model by running a Gibbs sampler for one million iterations. These samples are then used as a dataset from which we re-estimate the Ising model using Persistent Contrastive Divergence (PCD) (Neal, 1992; Younes, 1999; Tieleman, 2008; Du & Mordatch, 2019), which is an approximation to gradient-based maximum likelihood learning that requires an MCMC sampler; see details in Appendix L and pseudocode in Algorithm L.6.

PCD has two key free-parameters: the MCMC sampler itself and the number of sampling steps $K$ per parameter update. Better samplers enable lower values of $K$ to obtain a desired level of estimation error. We assess the performance of the samplers in terms of the estimation error of the estimated matrix $J$, as measured by the Frobenius norm $\|J - J^*\|_F$. The model's bias $\mathbf{b}$ is fixed at the ground-truth value.

We compare two types of PAVG: model-agnostic and model-specific. The former is the approach we used in previous experiments as discussed in Section 3.3.1. The model-specific version uses $J$ as the preconditioning matrix. As described at the end of 3.3, this renders PAVG identical to the log-quadratic Block-Gibbs sampler of Martens & Sutskever (2010).

### 5.3.1 Results

Table 1 shows the results. The model-specific PAVG method performs best, achieving low estimation error even when $K = 1$. This is not entirely surprising given the correspondence to the method by Martens & Sutskever (2010)

Table 2: Estimation error of quadratic term when sampling from a particular kind of deep energy-based model, with standard deviations across 5 runs in parentheses. Each run uses a different seed for all sources of randomness.

| | K=5* | K=10 | K=15 | K=20 |
|---|---|---|---|---|
| NCG | **0.128** (±.001) | **0.116** (±.0002) | **0.112** (±.0009) | **0.112** (±.0004) |
| AVG | 3.310 (±.078) | 2.599 (±.051) | 0.496 (±.119) | **0.114** (±.0006) |
| PAVG | 3.212 (±.079) | **0.117** (±.0004) | **0.114** (±.0006) | **0.112** (±.0007) |
| GWG | 0.732 (±.013) | 0.156 (±.003) | 0.120 (±.001) | **0.114** (±.0005) |
| Gibbs | 3.121 (±.146) | 2.270 (±.067) | 1.637 (±.007) | 1.204 (±.008) |

* K = number of MCMC steps for *AVG*; suitable multipliers ensure each method has the same budget.

that was developed for log-quadratic target distributions. However, the superiority of this model-specific sampler over GWG is a new finding; indeed Grathwohl et al. (2021) perform multiple experiments with log-quadratic target distributions but do not compare to Martens & Sutskever (2010).

Amongst the non model-specific methods, the results imply the following ranking: NCG > PAVG (model-agnostic) > GWG > Gibbs > AVG. In contrast to previous experiments, we now see a modest but clear advantage for NCG over PAVG, and that AVG underperforms compared to all other samplers unless $K$ is sufficiently large. It's important to note that, throughout learning, the $\epsilon$ step-size parameter used by NCG and (P)AVG is held fixed. This means that the same step-size must be effective across a *range* of target distributions (since the target changes every time we update the model's parameters). The results show that NCG and PAVG work robustly with a fixed step-size, whilst AVG is less robust. We suspect that all three methods would benefit from adaptively-tuning the step-size throughout learning, but leave this possibility to future work.

### 5.4 Sampling deep convolutional energy-based models

Deep energy-based models (EBMs) take the form $\log p(\mathbf{s}) = f(\mathbf{s}) - \log Z$, where $f$ is a deep neural network. Such models have attracted significant attention for continuous data (Du & Mordatch, 2019; Arbel et al., 2020; Qin et al., 2022) where Langevin-based samplers are the default approach. Less progress has been made in the discrete setting, with GWG (Grathwohl et al., 2021) being the first paper to showcase the potential of deep EBMs here.

A major challenge in comparing the efficacy of different samplers for deep EBMs is the lack of an easy-to-compute evaluation metric. Unlike the Ising model, parameter estimation error is not meaningful since the models are not identifiable. We propose a novel strategy for dealing with this problem consisting of the following steps:

**i)** Given a real dataset, fit a ground-truth energy-based model of the form $\frac{1}{2}\mathbf{s}^T J\mathbf{s} + f(\mathbf{s})$ where $f$ is a neural network and $J$ is symmetric. This fitting can be done via PCD with any reference MCMC sampler and a large number of sampling steps K (we use GWG and $K = 50$). Denote the final learned model by $\frac{1}{2}\mathbf{s}^T J^*\mathbf{s} + f^*(\mathbf{s})$.

**ii)** Sample a dataset $\mathcal{D}$ from the learned model by running the reference sampler for many (e.g. 50k) iterations.

**iii)** Fit a model of the form $\frac{1}{2}\mathbf{s}^T J\mathbf{s} + f^*(\mathbf{s})$ to the dataset $\mathcal{D}$, where $f^*$ is fixed, and only the symmetric matrix $J$ is estimated. This fitting is done with PCD, using the MCMC sampler we wish to assess. This model is identifiable, and hence the estimation error, $\|J - J^*\|_F$ is a valid performance metric.

We apply this methodology to the USPS 256-dimensional image dataset of binarised handwritten digits (Hull, 1994) and parameterise $f_\theta$ as a 7-layer convolutional network as detailed in Appendix O.

### 5.4.1 Results

Table 2 shows the results. Roughly speaking, we can rank the methods as NCG > PAVG ≥ GWG > AVG ≥ Gibbs. This ranking is similar that obtained in the previous experiment, and demonstrates that the newly proposed samplers are still effective for rather high-dimensional and non-linear target distributions. We note that the estimation errors in Table 2 correlate well with sample-based metrics like MMD and visual sample quality; see Figures 13 and 15 in the appendix.

## 6    Discussion

We have presented multiple discrete gradient-based MCMC samplers that show strong performance across a range of problem types in Bayesian inference and energy-based modelling. In particular, we obtained the NCG sampler by viewing the Metropolis-adjusted Langevin Algorithm through the lens of locally-informed proposals (Zanella, 2020) and the PAVG sampler through the lens of gradient-based auxiliary samplers Titsias & Papaspiliopoulos (2018).

Depending on the task, we saw that either NCG or PAVG show the strongest performance, and both generally outperform Gibbs-with-Gradients (GWG) by a clear margin. This demonstrates the value of using proposal distributions that update multiple dimensions at once, and do so in a correlated way. However, it is important to note that these advantages do not come for free: GWG requires no tuning, whereas both NCG and PAVG have step-size parameters. In practice, we would recommend running GWG alongside the new methods when facing a new sampling problem, as it provides a strong and reliable baseline.

To our knowledge, we are the first to adapt the auxiliary variable framework of Titsias & Papaspiliopoulos (2018) to discrete state-spaces. We believe there is significant scope to build on this idea, by investigating different choices of continuous conditional distributions, and understanding how/when *marginalised* auxiliary proposals can be leveraged. In the context of PAVG, a natural question is what happens if one replaces the *global* preconditioning matrix with a state-specific matrix such as the Hessian, in a similar vein to Manifold MALA (Girolami & Calderhead, 2011).

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

## A  One-hot categorical random variables

A categorical random variable with $k$ possible values can be represented as a one-hot vector belonging to the set

$$\mathcal{S} = \{\mathbf{s} : \mathbf{s}_i \in \{0,1\}, \sum_{i=1}^{k} s_i = 1\} \subset \{0,1\}^k \qquad \text{where} \;\; |\mathcal{S}| = k \qquad (24)$$

A $d$-length vector of $k$-valued categorical random variables thus belongs to the set $\mathcal{S}^d \subset \mathbb{R}^{dk}$.

The methods introduced in this paper (NCG, AVG and PAVG) all use proposal distributions that, before normalisation, have the form $\exp(\boldsymbol{a}^T \mathbf{s} + \boldsymbol{b}^T (\mathbf{s} \odot \mathbf{s}))$ for some choice of $\boldsymbol{a}$ and $\boldsymbol{b}$. Such formulas remain valid in the categorical case. However, the *normalised* versions of these distributions have a slightly different form. Specifically, to normalise this distribution, we note that there are $d$-independent groups of dimensions, and that within each group, there are only $k$ possible settings, yielding

$$\prod_{j=1}^{d} \sigma(\boldsymbol{a}_j^T \mathbf{s}_j + \boldsymbol{b}_j^T(\mathbf{s}_j \odot \mathbf{s}_j)) \qquad\qquad \sigma(\mathrm{x}) = \frac{\exp(\mathrm{x})}{\sum_{\mathcal{S}} \exp(\mathrm{x})}, \qquad (25)$$

where $\mathbf{s}_j := \mathbf{s}_{(j-1)k:jk}$, $\boldsymbol{a}_j := \boldsymbol{a}_{(j-1)k:jk}$ and $\boldsymbol{b}_j := \boldsymbol{b}_{(j-1)k:jk}$.

## B  Sampling algorithms

---

**Algorithm B.1** NCG step

---

**Require:** Step-size $\epsilon$. Unnormalised log prob function $f(\cdot)$. Triple $(\mathbf{s}_t, f(\mathbf{s}_t), \nabla f(\mathbf{s}_t))$.

  Sample $\mathbf{s}_{t+1} \sim q_\epsilon(\mathbf{s} \mid \mathbf{s}_t)$ as in Eq. 8 (binary/ordinal) or the categorical equivalent implied by Eq. 25

  Compute $f(\mathbf{s}_{t+1})$ & $\nabla f(\mathbf{s}_{t+1})$

  Accept $\mathbf{s}_{t+1}$ with probability $\quad \min\left(1, \exp(f(\mathbf{s}_{t+1}) - f(\mathbf{s}_t)) \dfrac{q_\epsilon(\mathbf{s}_t \mid \mathbf{s}_{t+1})}{q_\epsilon(\mathbf{s}_{t+1} \mid \mathbf{s}_t)}\right)$     (26)

---

---

**Algorithm B.2** AVG step

---

**Require:** Step-size $\epsilon$. Unnormalised log prob function $f(\cdot)$. Triple $(\mathbf{s}_t, f(\mathbf{s}_t), \nabla f(\mathbf{s}_t))$.

  Sample $\mathbf{z}_t \sim \mathcal{N}(\mathbf{z}; \; \sqrt{2/\epsilon} \, \mathbf{s}_t)$

  Sample $\mathbf{s}_{t+1} \sim q_\epsilon(\mathbf{s} \mid \mathbf{z}_t, \mathbf{s}_t)$ as in Eq. 13 (binary/ordinal) or the categorical equivalent implied by Eq. 25

  Compute $f(\mathbf{s}_{t+1})$ & $\nabla f(\mathbf{s}_{t+1})$

  Accept $\mathbf{s}_{t+1}$ with probability

$$\min\left(1, \exp(f(\mathbf{s}_{t+1}) - f(\mathbf{s}_t)) \frac{\mathcal{N}(\mathbf{z}; \; \sqrt{2/\epsilon} \, \mathbf{s}_{t+1}, \boldsymbol{I})}{\mathcal{N}(\mathbf{z}; \; \sqrt{2/\epsilon} \, \mathbf{s}_t, \boldsymbol{I})} \frac{q_\epsilon(\mathbf{s}_t \mid \mathbf{z}_t, \mathbf{s}_{t+1})}{q_\epsilon(\mathbf{s}_{t+1} \mid \mathbf{z}_t, \mathbf{s}_t)}\right) \qquad (27)$$

---

---

**Algorithm B.3** PAVG step

---

**Require:** Step-size $\epsilon$. Preconditioner $M$. Unnormalised log prob function $f(\cdot)$. Triple $(\mathbf{s}_t, f(\mathbf{s}_t), \nabla f(\mathbf{s}_t))$.

  Sample $\mathbf{z}_t \sim \mathcal{N}(\mathbf{z}; \; M_\epsilon^{1/2} \mathbf{s}_t, \boldsymbol{I})$ for $M_\epsilon$ defined in Eq. 39.

  Sample $\mathbf{s}_{t+1} \sim q_\epsilon(\mathbf{s} \mid \mathbf{z}_t, \mathbf{s}_t)$ as in Eq. 40 (binary/ordinal) or the categorical equivalent implied by Eq. 25

  Compute $f(\mathbf{s}_{t+1})$ & $\nabla f(\mathbf{s}_{t+1})$

  Accept $\mathbf{s}_{t+1}$ with probability

$$\min\left(1, \exp(f(\mathbf{s}_{t+1}) - f(\mathbf{s}_t)) \frac{\mathcal{N}(\mathbf{z}; \; M_\epsilon^{1/2} \mathbf{s}_{t+1}, \boldsymbol{I})}{\mathcal{N}(\mathbf{z}; \; M_\epsilon^{1/2} \mathbf{s}_t, \boldsymbol{I})} \frac{q_\epsilon(\mathbf{s}_t \mid \mathbf{z}_t, \mathbf{s}_{t+1})}{q_\epsilon(\mathbf{s}_{t+1} \mid \mathbf{z}_t, \mathbf{s}_t)}\right) \qquad (28)$$

---

Algorithms B.1, B.2 and B.3 show how to perform a single step of NCG, AVG and PAVG, respectively. For simplicity of presentation, we write these algorithms for a single MCMC chain. However, it is straightforward to use a vectorised implementation for a batch of MCMC chains, which is what we do in practice since it enables efficient GPU acceleration. This vectorised `PyTorch` code will be accessible upon publication.

## C  The computational benefit of NCG over GWG with large Hamming balls

We analyse the computational cost of GWG for Hamming balls of radius $r$ (henceforth GWG-$r$) and contrast this with the cost of NCG. For simplicity, we assume we are working in a binary space, $\mathcal{S}^d = \{0,1\}^d$, but similar arguments hold in the ordinal/categorical cases. The proposal distribution for GWG-$r$ is

$$q(\mathbf{s} \mid \mathbf{s}_t) = \frac{\exp\left(\frac{1}{2}\nabla f(\mathbf{s}_t)^T(\mathbf{s}-\mathbf{s}_t)\right)\mathbb{I}(\mathbf{s} \in H_r(\mathbf{s}_t))}{Z(\mathbf{s}_t)} \tag{29}$$

where $Z(\mathbf{s}_t) := \sum_{\mathbf{s}\in H_r(\mathbf{s}_t)} \exp\left(\frac{1}{2}\nabla f(\mathbf{s}_t)^T(\mathbf{s}-\mathbf{s}_t)\right)$ and $\mathcal{H}_r(\mathbf{s}_t)$ contains all those points in $\mathcal{S}^d$ that differ from $\mathbf{s}_t$ in *at most* $r$ dimensions. A naïve computation of $Z(\mathbf{s}_t)$ involves enumerating every point in the Hamming ball $H_r(\mathbf{s}_t)$; the number of terms in the sum is $\sum_{k=0}^{r}\binom{d}{k}$, which grows extremely quickly as a function of $r$ as illustrated in Table 3. It is for this reason that we claim in the main text that a 'straightforward approach' to using large values of $r$ is prohibitively expensive.

Table 3: Growth of size of Hamming ball as function of dimension $d$ and radius $r$

|  | $r=1$ | $r=2$ | $r=3$ | $r=4$ | $r=5$ | $r=6$ |
|---|---|---|---|---|---|---|
| $d=50$ | 51 | 1276 | 20876 | 251176 | 2369936 | 18260636 |
| $d=100$ | 101 | 5051 | 166751 | 4087976 | 79375496 | 1271427896 |
| $d=200$ | 201 | 20101 | 1333501 | 66018451 | 2601668491 | 85010294791 |

We believe it should be possible to compute $Z(\mathbf{s}_t)$ more efficiently through suitable rearrangements and caching of computations in the manner of dynamic programming, however this is far from straightforward to implement, especially when working with vectorised batches of MCMC chains. More fundamentally, such a method for computing $Z(\mathbf{s}_t)$ does not immediately solve the issue of sampling the proposal in Equation 29. The naïve way for sampling this proposal is to enumerate every point $\mathbf{s}$ in the Hamming ball along with their probabilities $\exp\left(\frac{1}{2}\nabla f(\mathbf{s}_t)^T(\mathbf{s}-\mathbf{s}_t)\right)/Z(\mathbf{s}_t)$, and then use the fact these probabilities define a categorical distribution which can be sampled in the usual manner (i.e. inverse transform sampling). Enumerating all $\sum_{k=0}^{r}\binom{d}{k}$ probabilities is prohibitively expensive for large values of $r$, and (to the best of our knowledge) it is an open problem if/how the dynamic programming approach mentioned above can be leveraged to reduce this sampling cost.

In contrast, NCG uses the *fully-factorised* proposal distribution that can be written as

$$q_\epsilon(\mathbf{s} \mid \mathbf{s}_t) = \prod_{i=1}^{d} \frac{1}{Z_i(\mathbf{s}_t)} \exp\left(\left[\frac{1}{2}\nabla f(\mathbf{s}_t)_i + \frac{1}{\epsilon}\mathbf{s}_{t,i}\right]\mathbf{s}_i - \frac{1}{2\epsilon}\mathbf{s}_i^2\right) \tag{30}$$

where $Z_i(\mathbf{s}_t) := \sum_{\mathcal{S}} \exp\left(\left[\frac{1}{2}\nabla f(\mathbf{s}_t)_i + \frac{1}{\epsilon}\mathbf{s}_{t,i}\right]\mathbf{s}_i - \frac{1}{2\epsilon}\mathbf{s}_i^2\right)$. Each per-dimension normaliser $Z_i(\mathbf{s}_t)$ is just a sum over $|\mathcal{S}|$ terms (i.e. 2 terms for binary variables), and since there are only $d$ such normalisers, the total number of terms that need computing is only $d|\mathcal{S}|$. Moreover, sampling the NCG proposal distribution is also easy since it is fully factorised, and hence we can sample each dimension independently. For instance, when working with binary vectors, all we need to do is sample $d$ independent Bernoulli distributions. We note that the other proposed samplers in this work (AVG & PAVG) also use *fully-factorised* proposal distributions, and hence the same arguments we just provided for NCG apply to them as well.

Finally, we note that GWG (with radius 1), NCG, AVG and PAVG all have very similar run-times in practice (see Section I) since the predominant cost in all cases is a single function and gradient evaluation ($f(\mathbf{s}_t), \nabla f(\mathbf{s}_t)$) per iteration.

# D    Interpretation of auxiliary variables

We aim here to shed some light on the role of the auxiliary variables used in Sections 3.2 and 3.3. To keep things simple, we illustrate how AVG works for a single binary variables $s$ with target distribution $p(s) = \exp(f(s))/Z$. In this simple scenario, $Z$ can be calculated as $\exp(f(0)) + \exp(f(1))$, and hence $p(s)$ is a known function. Recall that the first step in AVG is augmenting this binary variable with Gaussian auxiliary variables distributed according to $p(z \mid s) = \mathcal{N}(z; \sqrt{\frac{2}{\epsilon}}s, 1)$. Figure 5 plots these conditional Gaussian distributions for two different values of $\epsilon$.

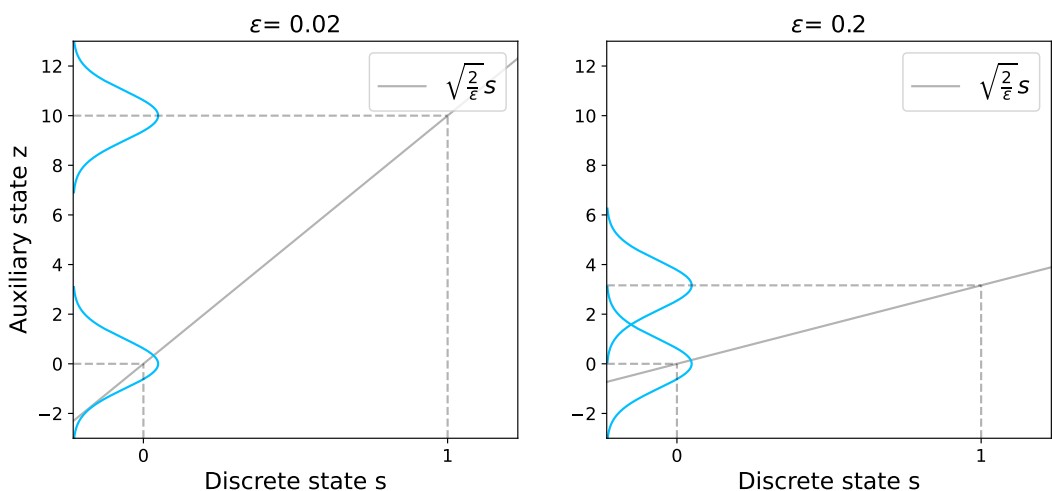

Figure 5: Conditional Gaussian distributions used in AVG for two different values of $\epsilon$.

The key feature of these plots is that $\epsilon$ controls the amount of separation between the two Gaussian distributions (for $s = 0$ and $s = 1$). Greater separation means that a block-wise MCMC sampler is less likely to move between the two binary states. Why is this the case? An informal argument goes as follows: imagine we are using $\epsilon = 0.02$ (left plot) and our current state is $s = 0$. Conditional on $s = 0$, we sample a $z$, i.e. we sample from the Gaussian shown at the bottom of the y-axis. Suppose we sample $z = 2$. Now, we would like to sample an $s$ conditional on $z = 2$, i.e. we would like to sample a value of $s$ that could have plausibly generated $z = 2$. It is very unlikely that $s = 1$ could have generated $z = 2$ since that would be an 8-sigma event (the Gaussian at the top of y-axis has mean 10 and standard deviation 1). With $s = 1$ excluded, we therefore end up sampling $s = 0$, and remain stuck there.

The above analysis is imprecise in a couple of ways. We were intentionally vague on how to sample $s$ given $z$. As explained in the main paper, AVG uses an MH accept-reject step with a particular proposal distribution. Here, we will simplify things and use the exact conditional $p(s \mid z)$, which is given by Bayes rule

$$p(s \mid z) = \frac{p(s)\,\mathcal{N}(z;\ \sqrt{\frac{2}{\epsilon}}s, 1)}{p(0)\,\mathcal{N}(z;\ 0, 1) + p(1)\,\mathcal{N}(z;\ \sqrt{\frac{2}{\epsilon}}, 1)}. \tag{31}$$

With this distribution at hand, we can be more precise about what happens when we sample from $p(s \mid z = 2)$. Probability mass will concentrate on $s = 0$ (as we claimed before) *unless* the 'prior probability' $p(1)$ is extremely close to 1. Thus, for 'most' target distributions (i.e. those that are not nearly point-masses), our block-wise MH algorithm will be 'sticky', remaining at $s = 0$ for an extraordinarily long period of time.

The analysis changes however when we use the larger step-size of $\epsilon = 0.2$ (right subplot). Now, if we repeat our argument, and consider starting at $s = 0$ and sampling $z = 2$, there is a much greater chance that the next state in the chain is $s = 1$. This is because $z = 2$ is a plausible value under the Gaussian distribution corresponding to $s = 1$ (in fact, more plausible than the Gaussian for $s = 0$). Our block-wise MH algorithm will be less sticky than before, and we will see faster convergence to the target distribution.

If larger values of $\epsilon$ are advantageous, why not use an extremely large value? This actually is the correct thing to do if we can perform exact block-Gibbs sampling and we don't require an MH step. This is because as $\epsilon \to \infty$,

$s$ and $z$ become independent (all values of $s$ map to the same Gaussian distribution $\mathcal{N}(0,1)$) and hence we have $p(s \,|\, z) = p(s)$. Thus, our exact block-sampler alternates sampling irrelevant Gaussian noise and exact samples from $p(s)$. However, this extreme scenario illustrates precisely why exact block-Gibbs sampling is generally infeasible since a special case of it ($\epsilon \to \infty$) assumes we can evaluate and sample $p(s)$, which is precisely the problem we are trying to solve in the first place! In practice, we cannot evaluate or sample $p(s \,|\, z)$, and we need to approximate it using a proposal distribution. Our choice of proposal distribution for AVG (based on a local first-order Taylor approximation) is often reasonable for sufficiently small values of $\epsilon$, but may be poor for larger values, resulting in unacceptably high rejection rates in the MH accept-reject step.

## E   Preconditioned MALA as an auxiliary variable scheme

Here, we explain how to frame PMALA (equation 4) as an auxiliary variable sampler, and why its 'discrete analogue' is intractable. Our derivation here has the same structure as that of AVG in Section 3.2, except we now use a different form for the conditional Gaussian auxiliary variables.

For a continuous state $\mathbf{s} \in \mathbb{R}^d$, consider the unnormalised target density $\pi(\mathbf{s}, \mathbf{z}) = \exp(f(\mathbf{s}))\mathcal{N}(\mathbf{z}; M^{-1/2}\mathbf{s}, \boldsymbol{I})$. In theory, this distribution could be sampled in a block-Gibbs fashion via alternate sampling of $\mathbf{z}_t \sim \mathcal{N}(\mathbf{z}; M^{-1/2}\mathbf{s}, \boldsymbol{I})$, and $\mathbf{s}_{t+1} \sim \pi(\mathbf{s} \,|\, \mathbf{z}_t) \propto \pi(\mathbf{s}, \mathbf{z}_t)$. However, for general functions $f$ this second sampling step is intractable, so it is replaced with an MH accept-reject step using the proposal distribution

$$q_\epsilon(\mathbf{s} \,|\, \mathbf{z}_t, \mathbf{s}_t) \propto \exp(f(\mathbf{s}_t) + \nabla f(\mathbf{s}_t)^T(\mathbf{s} - \mathbf{s}_t))\mathcal{N}(\mathbf{z}_t; M^{-1/2}\mathbf{s}, \boldsymbol{I}) \tag{32}$$

$$\propto \exp\left(-\frac{1}{2}(\mathbf{s} - \boldsymbol{\mu})^T M^{-1}(\mathbf{s} - \boldsymbol{\mu})\right) \qquad \text{where} \quad \boldsymbol{\mu} := M^{1/2}\mathbf{z}_t + M\nabla f(\mathbf{s}_t) \tag{33}$$

$$= \mathcal{N}(\mathbf{s}; \; M^{1/2}\mathbf{z}_t + M\nabla f(\mathbf{s}_t), M) \tag{34}$$

where equation 32 approximates $\pi(\mathbf{s}, \mathbf{z}_t)$ via a Taylor expansion of $f(\mathbf{s})$. If we now marginalise out the latents, we obtain

$$\int \mathcal{N}(\mathbf{z}_t; M^{-\frac{1}{2}}\mathbf{s}_t, \boldsymbol{I})\mathcal{N}(\mathbf{s}; \; M^{1/2}\mathbf{z}_t + M\nabla f(\mathbf{s}_t), M)d\mathbf{z}_t = \mathcal{N}(\mathbf{s}; \; \mathbf{s}_t + M\nabla f(\mathbf{s}_t), 2M) \tag{35}$$

which is equal to the PMALA proposal in equation 4 after redefining $M$ as $\frac{\epsilon}{2}M$.

If we now restrict $\mathbf{s} \in \mathcal{S}^d \subset \mathbb{R}^d$ to be a discrete random variable in equation 33, then we no longer obtain a Gaussian proposal distribution as in equation 34, but rather a discrete pairwise Markov random field that is generally intractable to normalise and sample.

## F   Choice of preconditioning matrix for PAVG

We use matrices of the form $\gamma M$, where $\gamma$ is an adaptively learned scaling parameter. In general settings, when we have no domain-knowledge to help us select $M$, we choose it to be an empirical covariance/precision matrix (see below for how to choose) computed from a set of initial samples obtained during the burn-in phase. The algorithm for this general case is presented as algorithm F.4.

However, when sampling from energy-based models (EBMs), we usually have access to a real-world dataset that the EBM is modelling. In this case, we can compute the empirical covariance/precision of this dataset, and use that as our $M$. This simplifies algorithm F.4: we drop lines 5-8, and no longer need a zero-initialisation of $M$ in line 2.

### F.1   Covariance or precision? Automatically picking $M$ from a list of options.

If we were sampling continuous variables drawn from a Gaussian distribution, then the correct choice of $M$, based on the approximation we made in equation 14, would be the (negative) precision matrix. This choice also works best in our ordinal experiments in Section 5.1. However, in all of our binary experiments, the *covariance* matrix, not precision, works better (often substantially so).

The choice between covariance and precision (or, more generally, a list of candidate matrices) is a hyperparameter. To avoid manual tuning, we propose a simple heuristic to automatically select the 'best' matrix from a list of

options. We define 'best' as the matrix that (after rescaling) minimises the approximation error in equation 14 across pairs of consecutive samples accumulated during a burn-in phase. Specifically, we do the following:

1. During the burn-in period, collect $(\mathbf{s}_t, f(\mathbf{s}_t), \nabla f(\mathbf{s}_t))$ into a 'dataset' $\mathcal{D}$.

2. Assemble a list of candidates $[M_1, M_2 \dots M_m]$. In our experiments, we consider only two candidates: the empirical covariance & precision matrix of collected samples $\{\mathbf{s}_t\}$.

3. For each $M$ in our list, solve the least-squares linear regression problem for the *scalar* $\gamma_0$

$$\arg\min_{\gamma_0} \sum_t \|y_t - \gamma_0 x_t\|^2 \tag{36}$$

$$\text{where} \quad y_t := f(\mathbf{s}_{t+1}) - f(\mathbf{s}_t) - \nabla f(\mathbf{s}_t)^T (\mathbf{s}_{t+1} - \mathbf{s}_t), \quad x_t := (1/2)(\mathbf{s}_{t+1} - \mathbf{s}_t)^T M (\mathbf{s}_{t+1} - \mathbf{s}_t) \tag{37}$$

4. Return the re-scaled matrix $\gamma_0 M$ which obtained the lowest least-squared loss.

Across many of our experiments, the returned matrix $\gamma_0 M$ performed very well without any additional alterations. However, to maximise the performance of PAVG, we found it beneficial to only use the scaling factor $\gamma_0$ as an 'initialisation' and continue to update it adaptively using `AdaptGamma` as shown in Algorithms F.4 and F.5.

### F.2   Preconditioners with negative eigenvalues

When deriving PAVG in section 3.3, we defined a conditional Gaussian distribution of the form

$$\mathcal{N}(\mathbf{z}; \ M_\epsilon^{1/2}\mathbf{s}_t, \boldsymbol{I}), \qquad\qquad M_\epsilon := M + (2/\epsilon)\boldsymbol{I}. \tag{38}$$

We note that there are multiple types of matrix square-root, and any of them is valid here. However, such square-roots only exist if $M_\epsilon$ is semi-positive definite (i.e. all eigenvalues are non-negative). This is only guaranteed to be the case (for any value of $\epsilon$) if $M$ is also semi-positive. This is because the eigenvalues of $M_\epsilon$ are of the form $\lambda_i + (2/\epsilon)$, where $\lambda_i$ is an eigenvalue of $M$.

We can specify an alternative definition of $M_\epsilon$ that ensures semi-positive definiteness

$$M_\epsilon := M + \underbrace{\left[\max(0, -\lambda_{\min}) + (2/\epsilon)\right]}_{:= d_\epsilon} \boldsymbol{I} \tag{39}$$

where $\lambda_{\min} \leq \lambda_i$ for all $i$. This new definition of $M_\epsilon$ means that PAVG uses a different conditional Gaussian, and the MH-proposal distribution in equation 17 now becomes

$$q_\epsilon(\mathbf{s} \mid \mathbf{z}_t, \mathbf{s}_t) = \prod_{i=1}^n \sigma\left(\left[\nabla f(\mathbf{s}_t)_i - (M\mathbf{s}_t)_i + (M_\epsilon^{1/2}\mathbf{z}_t)_i\right]\mathbf{s}_i - \frac{d_\epsilon}{2}\mathbf{s}_i^2\right), \qquad \text{where} \quad \sigma(\mathbf{x}) = \frac{\exp(\mathbf{x})}{\sum_{\mathbf{x}\in\mathcal{S}}\exp(\mathbf{x})}, \tag{40}$$

What is the consequence of this new definition of $M_\epsilon$? Previously, we diagonally perturbed $M$ by some value in $(0, \infty)$, where larger step-sizes $\epsilon$ meant smaller perturbations. Now, we diagonally perturb $M$ by some value in $(d_\infty, \infty)$, where $d_\infty := \max(0, -\lambda_{\min}) \geq 0$. Thus compared to the old regime, the new regime enforces a kind of maximum step-size (minimum perturbation).

## G   Baseline ordinal samplers

Our ordinal experiments use two additional baselines: Ordinal-GWG and MH-uniform. Ordinal-GWG is newly introduced in this paper as it is arguably an 'obvious' way to improve GWG when dealing with ordinal data, and incurs very little additional implementation or computational complexity compared to standard GWG. We provide visual illustrations of the proposal distributions of these Metropolis-Hastings samplers in Figure 6. As can be seen from the figure, Ordinal-GWG, like GWG, can only update *one* dimension at a time. However, it can jump multiple states along any given dimension, with the maximum number of moves controlled by a radius parameter $r$. Thus,

---

**Algorithm F.4** Adaptive learning of preconditioning matrix. (Default values in brackets are used across *all* experiments)

---

**Require:** Integers $N_{\text{iters}}$, $N_{\text{chains}}$ (100), $N_M$ (1000) and $N_{\text{adapt}}$ (100).

**Require:** Initial adaptation rate $\delta$ (0.25) and decay factor $\rho$ (0.99).

 1: Initialise chains $S_1 = [\mathbf{s}^1, \ldots, \mathbf{s}^{N_{\text{chains}}}]$ and history $\mathcal{D} = [S_1]$
 2: Initialise $M = 0$, $\gamma = \gamma_{\text{old}} = 1.0$
 3: **for** $t \in \{1, \ldots N_{\text{iters}}\}$ **do**
 4:      Compute $S_{t+1}$ from $S_t$ using one step of PAVG (Algorithm B.3) with preconditioner $\gamma M$
 5:      **if** $t < N_M$ **then**
 6:          Append $S_{t+1}$ to history $\mathcal{D}$
 7:      **else if** $t = N_M$ **then**
 8:          Define new $M$ as described in Section (F.1) using history $\mathcal{D}$
 9:      **else if** $t \mod N_{\text{adapt}} = 0$ **then**
10:          $\gamma, \gamma_{\text{old}} = \texttt{AdaptGamma}(\gamma, \gamma_{\text{old}}, \delta, N_{\text{adapt}}, \mathcal{D})$          ▷ Update scaling factor
11:          $\delta \leftarrow \rho\delta$          ▷ exponential decay of step-size
12:      **end if**
13: **end for**

---

**Algorithm F.5** Adapt $\gamma$ to maximise 'jump' distance $\|\mathbf{s}_t - \mathbf{s}_{t-1}\|_1$

---

   **function** ADAPTGAMMA($\gamma$, $\gamma_{\text{old}}$, $\delta$, $N_{\text{adapt}}$, $\mathcal{D}$)
     $a_{\text{new}} \leftarrow$ Average value of $\|\mathbf{s}_t - \mathbf{s}_{t-1}\|_1$ computed over all chains in $\mathcal{D}[-N_{\text{adapt}} :]$
     $a_{\text{old}} \leftarrow$ Average value of $\|\mathbf{s}_t - \mathbf{s}_{t-1}\|_1$ computed over all chains in $\mathcal{D}[-2N_{\text{adapt}} : -N_{\text{adapt}}]$
     $\texttt{Increased} \leftarrow \gamma \geq \gamma_{\text{old}}$
     $\texttt{Improved} \leftarrow a_{\text{new}} \geq a_{\text{old}}$
     **if** (**Increased and Improved**) **or** (**not Increased and not Improved**) **then**
         $\hat{\delta} \leftarrow \delta$          ▷ Positive adjustment
     **else**
         $\hat{\delta} \leftarrow -\delta$          ▷ Negative adjustment
     **end if**
     $\gamma_{\text{old}} \leftarrow \gamma$
     **if** $|\gamma| \geq 1$ **then**
         $\gamma \leftarrow \gamma * (1 + \hat{\delta})$          ▷ Multiplicative adjustment
     **else**
         $\gamma \leftarrow \gamma + \hat{\delta}$          ▷ Additive adjustment (allows $\gamma$ to change sign)
     **end if**
     **return** $\gamma, \gamma_{\text{old}}$
   **end function**

---

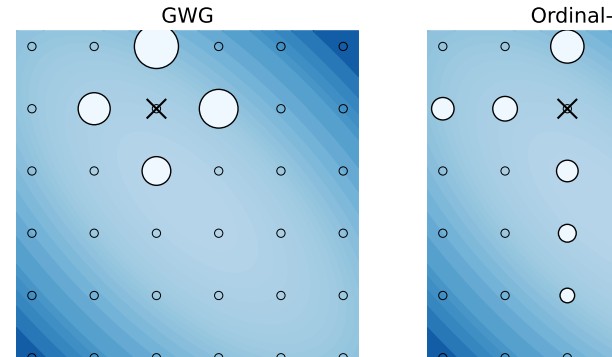
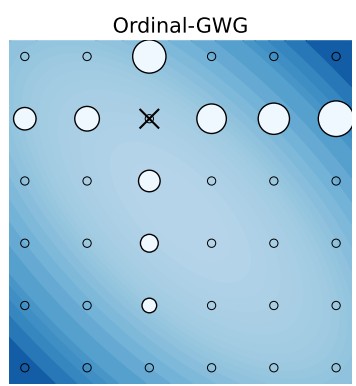
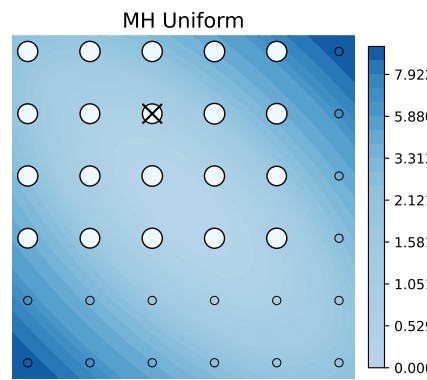

Figure 6: Illustration of the different proposal distributions used by various baseline MH samplers. The darker blue contours represent the target distribution, the white dots represent the proposal distribution around the current state (black X). **Left**: GWG **Middle:** Ordinal-GWG with radius 3 **Right:** Uniform MH with radius 2.

the support set of this proposal distribution is (at most) of size $(2r - 1)d$, where $d$ is the dimensionality. The Ordinal-GWG proposal has the same functional form as GWG in equation 6, except that the summation is no longer over a Hamming ball of radius 1, but instead over the aforementioned support set.

MH-uniform (final column of Figure 6) also has a radius parameter $r$. This radius controls the size of the hypercube over which the proposal is uniform.

## H    Tuning step-size parameters

NCG, AVG & PAVG have step-size parameters $\epsilon$. Our grid-search based tuning procedure involves running each sampler for a short amount of time (1000 iterations, which takes $\leq 1$ minute in most of our experiments) with different step-sizes, and selecting the step-size that maximises the average L1-distance $\|\mathbf{s}_{t+1} - \mathbf{s}_t\|_1$ between successive states (averaged over all time-steps and parallel chains). For NCG, AVG & PAVG we first first identify the best order-of-magnitude by searching, in parallel, over the 5 values in the set $\{0.05, 0.5, 5.0, 50.0, 500.0\}$. We did not find it necessary to consider other orders of magnitudes. Indeed, our analysis in Section D indicates that (at least in the context of AVG and binary variables) step-sizes below $\sim 0.01$ are unlikely to work. After identifying the best order-of-magnitude, we then search, in parallel, each decile within that particular order of magnitude e.g. $\{0.1, 0.2 \ldots, 0.9\}$. Thus, in total, our search procedure tests 15 values of $\epsilon$ and, due to the parallelism, only takes a couple of minutes for our experiments. Admittedly, for more complex experiments with expensive target distributions, the cost of this search procedure will increase.

Ordinal-GWG and MH-uniform also have step-size-like parameters $r$ and we tune these in a similar way as described above. We first search $r \in \{1, 5, 10, 15, 20\}$ and then search within the best interval e.g. $r \in \{15, 16, 17, 18, 19\}$ (note: the maximum possible value of the radius parameter is 50 in these experiments). The final selected step-size parameters are shown in Table 4.

It is natural to wonder how our selection of $\epsilon$ in PAVG interacts with the choice of preconditioning matrix. As explained in Appendix F, PAVG's preconditioning matrix is automatically tuned during *each* run. Hence, when we perform the above grid-search, we are fixing a particular $\epsilon$ at the beginning of a run, and then automatically learning the preconditioning matrix given that fixed $\epsilon$.

### H.1    Step-size sensitivity analysis

It is important to have a sense of how robust NCG, AVG and PAVG are to the choice of step-size. We investigate this in the context of the Ising model experiment in Section 5.3. We fix the number of sampling steps $K = 10$, and consider how the errors reported in Table 1 vary as we multiplicatively perturb the step-sizes used to generate those results. Doing so yields Figure 7. The main finding here is that all samplers are relatively stable over a wide range of step-sizes. For NCG and PAVG, we can multiply our original step-size by any number between 0.3 and

| Experiment | NCG | AVG | PAVG | Ordinal-GWG | MH-Uniform |
|---|---|---|---|---|---|
| Ordinal (poly2) | 0.05 | 0.02 | 1000.0 | 16 | 2 |
| Ordinal (poly4) | 0.05 | 0.02 | 0.06 | 8 | 1 |
| Bayesian regression | 0.03 | 1000.0 | 1000.0 | - | - |
| Ising lattice | 0.5 | 0.2 | 0.2 | - | - |
| ConvEBM | 0.2 | 0.1 | 0.1 | - | - |

Table 4: Final step-sizes used across different experiments after tuning.

10.0, and the resulting error is within $\sim 15\%$ of the original error and *remains lower* than GWG's error. Finally, we note that our chosen step-sizes (i.e. relative step-size $= 1$) are not optimal, which is to be expected since our search procedure described above does not have access to our error metric (which is important, because the error metric relies on ground-truth information that is not typically available for real-world problems).

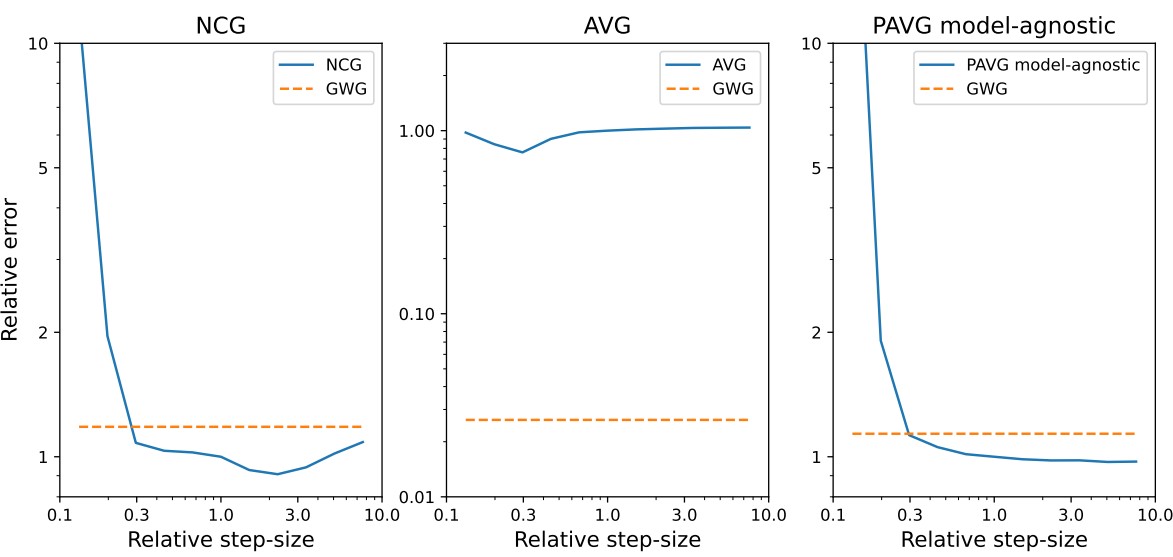

Figure 7: Relative error as we multiplicatively perturb the step-sizes used by NCG, AVG and PAVG in our Ising model experiments of 5.3. Relative error is defined as the new error divided by the 'original' error reported in Table 1. Similarly, the dashed line for GWG is also relative to the errors reported in Table 1.

# I  Wall-clock versus per-iteration results

In all the experiments presented in the main text, we allotted equal wall-clock time to each competing MCMC method. Whilst this is the fairest form of comparison, it is implementation and system-dependent. To give a sense of how the different methods compare on a per-iteration basis, see Figures 8 and 9 for 20D ordinal and 100D Bayesian regression results, respectively. Methods with lower wall-clock costs per-iteration are run for more iterations in total. In particular, the Gibbs sampler is run for significantly more iterations than other samplers due its low cost in these experiments. However, we note that Gibbs is not guaranteed to be the cheapest method universally; it can be very expensive for categorical distributions with large state-spaces (Grathwohl et al., 2021).

In Figure 8, GWG & GWG-ordinal run for the shortest number of iterations. This is potentially surprising since one may expect GWG to cost the same amount per-iteration as NCG. This occurs because GWG-based proposal distributions have irregularly shaped support sets when at the boundary of the state-space—see the centre panel

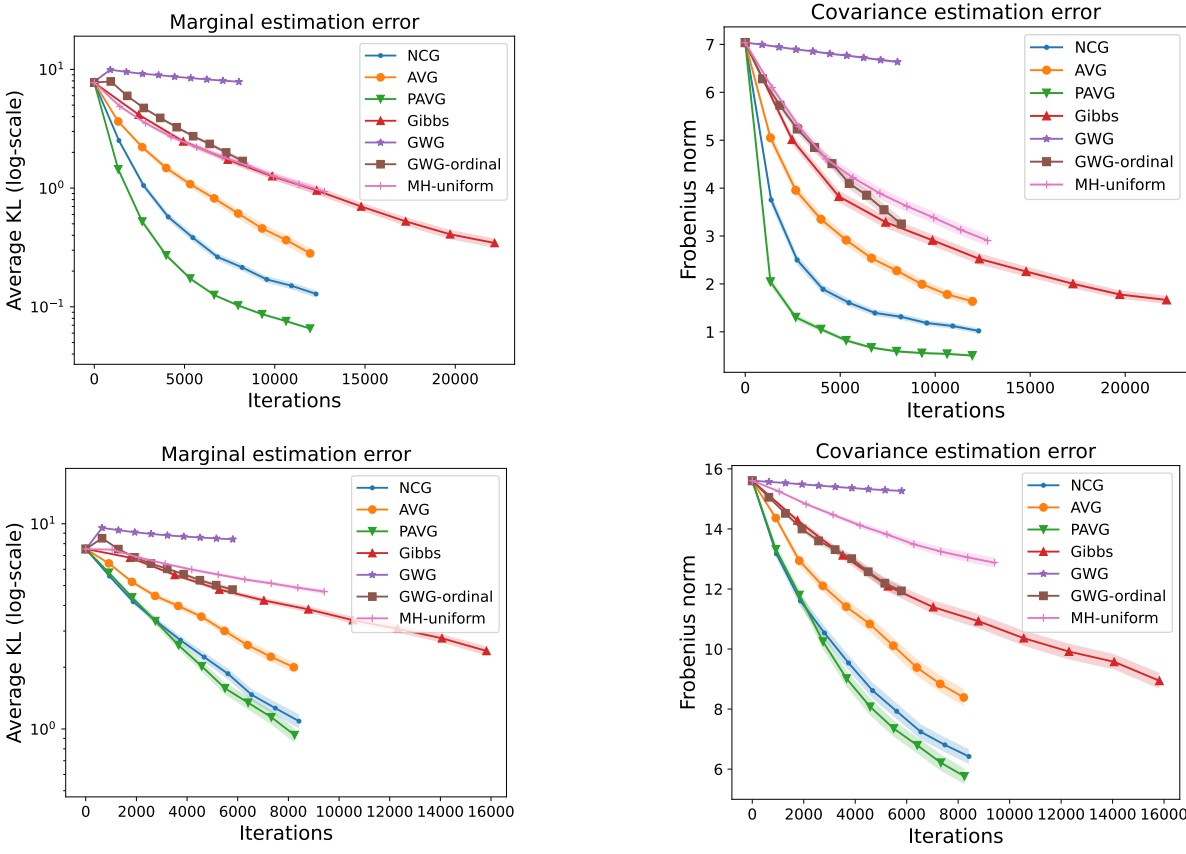

Figure 8: 20D mixture-of-polynomial results. These figures are the per-iteration versions of those in Figure 2.

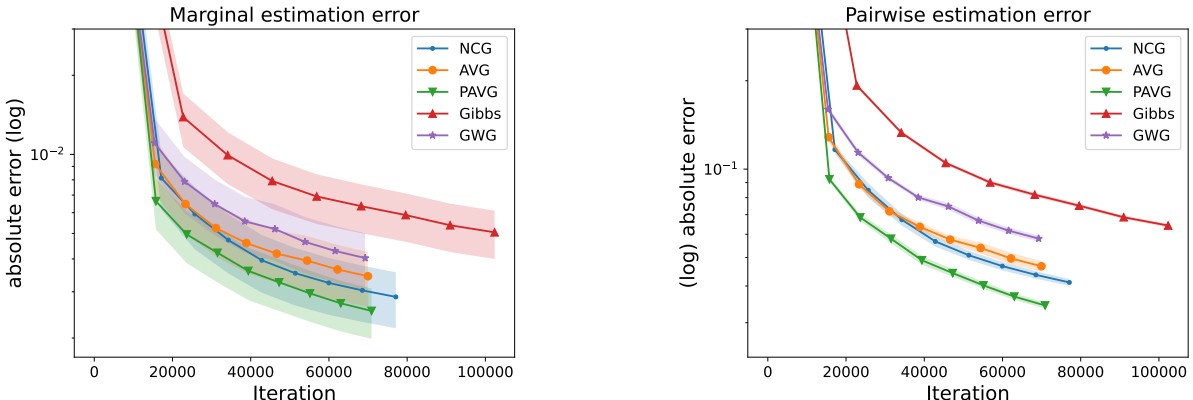

Figure 9: 100D Sparse Bayesian linear regression results. This Figure is the per-iteration version of Figure 4.

of Figure 6 for an illustration. Normalising the proposal over such irregularly shaped support sets is difficult to implement efficiently in a vectorised manner for a batch of MCMC chains (since each chain can have a differently-shaped support set, and complex masking operations are seemingly required to handle this). Even in an idealised scenario where this computational overhead is zero—and so GWG matches NCG in per-iteration cost—our rankings reported in the main text would not be impacted significantly.

## J    Evaluation metrics for ordinal & Bayesian regression experiments

The experiments in sections 5.1 and 5.2 use the same methodology for evaluating performance. In both cases we:

- Run 100 parallel chains for 10 minutes with a burn-in period of 1 minute. The resulting number of iterations per-method & experiment can be seen in Figures 8 and 9.

- After the burn-in period, we begin saving the history of each chain.

- Every minute after the burn-in, we use the chain histories to compute estimation errors (Table 5) for each chain *separately*. We then compute a mean and standard error *across* our 100 parallel chains, and use these to construct Figures 2 and 4.

- At the end of the run, we estimate the Effective Sample Size (ESS) of each chain. Following Zanella (2020) and Grathwohl et al. (2021), we map every state $\mathbf{s}$ in our chains to a test statistic, namely $\|\mathbf{s} - \mathbf{s}_{\text{random}}\|_1$, where $\mathbf{s}_{\text{random}}$ is a randomly selected point in the state-space. The resulting test statistics can be stored in a matrix $S$ of shape `num_iters` $\times 100$, and the ESS-per-chain is then estimated with `tfp.mcmc.effective_sample_size(S, filter_beyond_positive_pairs=True)` using version `0.14.1` of `tensorflow-probability`. Finally, to construct the ESS plots in Figures 2 and 4, we compute box-plot statistics (median & quartiles) across the 100 ESS estimates.

| Experiment | marginal error | covariance/pairwise error |
|---|---|---|
| Ordinal | $(1/d) \sum_{i=1}^{d} D_{KL}(q_i \parallel p_i)$ | $\|M_q - M_p\|_F$ |
| Bayesian regression | $(1/d) \sum_{i}^{d} \|q_i(\mathbf{s}_i = 1) - p_i(\mathbf{s}_i = 1)\|$ | $\frac{1}{d^2} \sum_{i,j}^{d} \sum_{k,l \in \{0,1\}} \|q_{i,j}(\mathbf{s}_i = k, \mathbf{s}_j = l) - p_{i,j}(\mathbf{s}_i = k, \mathbf{s}_j = l)\|$. |

Table 5: $q_i$ refers to the $i^{\text{th}}$ univariate marginal distribution of the empirical samples accumulated across a single chain. Similarly, $q_{i,j}$ refers to a bivariate marginal of such empirical samples. $p_i$ and $p_{i,j}$ refer to marginals of the target distribution, and are computed *exactly* (i.e. they are not empirical distributions). $M_q$ and $M_p$ are both empirical covariance matrices; the former is computed using samples accumulated across a single chain, whilst the latter is computed using $100,000$ samples drawn from the target distribution.

## K    Posterior distribution for sparse Bayesian linear regression

We define a Bayesian regression model using a similar procedure to Titsias & Yau (2017). The regression takes the form

$$\boldsymbol{y} = X(\mathbf{s} \odot \boldsymbol{\omega}) + \sigma \boldsymbol{\nu} \qquad\qquad \boldsymbol{y}, \boldsymbol{\nu} \in \mathbb{R}^N \quad \mathbf{s}, \boldsymbol{\omega} \in \mathbb{R}^D \qquad (41)$$

The quantities in this equation are random variables described by the following generative process

1. Place a sparsity-promoting prior on $\mathbf{s}$

$$p(\mathbf{s}) = \Gamma\big(\sum_{i=1}^{D} \mathbf{s}_i + \alpha_\pi\big)\Gamma\big(D - \sum_{i=1}^{D} \mathbf{s}_i + \beta_\pi\big), \qquad (42)$$

where $\Gamma$ is the gamma *function* (not distribution) and $(\alpha_\pi, \beta_\pi)$ are hyperparameters with default values $(0.001, 10.0)$. This prior can itself be viewed as the marginal of a Bayesian model specified by $\mathbf{s} \mid \pi \sim \prod_{i=1}^{D} Bernoulli(\mathbf{s}; \pi)$ and $\pi \sim Beta(\pi; \alpha_\pi, \beta_\pi)$.

2. Let $\boldsymbol{\nu} \sim \mathcal{N}(0, \boldsymbol{I}_N)$.

3. Define $X_{\mathbf{s}} = X \text{diag}(\mathbf{s})$ and note that $X(\mathbf{s} \odot \boldsymbol{\omega}) = X_{\mathbf{s}} \omega$.

4. Place a conjugate normal-inverse-gamma prior over weights and noise-variance $(\boldsymbol{\omega}, \sigma^2)$. Namely,

$$p(\boldsymbol{\omega}, \sigma^2 \mid \mathbf{s}, X) = \mathcal{N}(\boldsymbol{\omega} \mid 0, g\sigma^2(X_\mathbf{s}^T X_\mathbf{s} + \lambda \boldsymbol{I}_D)^{-1}) \; InvGamma(\sigma^2 \mid \alpha_\sigma, \beta_\sigma) \tag{43}$$

We note that this choice of normal distribution is a kind of perturbed g-prior (Zellner, 1986), with hyperpa-rameters $(g, \lambda)$ with default values $(20, 0.001)$. The inverse-gamma hyperparameters $(\alpha_\sigma, \beta_\sigma)$ have default values $(0.1, 0.1)$.

5. As implied by the regression formula in equation 41, our model of $\boldsymbol{y}$ follows

$$p(\boldsymbol{y} \mid \mathbf{s}, X, \boldsymbol{\omega}, \sigma^2) = \mathcal{N}(\boldsymbol{y} \mid X_\mathbf{s}\boldsymbol{\omega}, \sigma^2 \boldsymbol{I}_N) \tag{44}$$

6. Putting the previous steps together, we arrive at the joint distribution

$$p(\mathbf{s}, \sigma^2, \boldsymbol{\omega}, \boldsymbol{y} \mid X) = p(\mathbf{s})p(\boldsymbol{\omega}, \sigma^2 \mid \mathbf{s}, X)p(\boldsymbol{y} \mid \mathbf{s}, \boldsymbol{\omega}, \sigma^2, X) \tag{45}$$

The variables $(\sigma^2, \boldsymbol{\omega})$ can be analytically integrated out, resulting in the *posterior distribution*

$$p(\mathbf{s} \mid \boldsymbol{y}, X) \propto p(\mathbf{s}) \frac{|X_\mathbf{s}^T X_\mathbf{s} + \lambda \boldsymbol{I}_D|}{|(1+g)X_\mathbf{s}^T X_\mathbf{s} + \lambda \boldsymbol{I}_D|} \left(2\beta_\sigma + \boldsymbol{y}^T \boldsymbol{y} - g\boldsymbol{y}^T X_\mathbf{s} \left[(1+g)X_\mathbf{s}^T X_\mathbf{s} + \lambda \boldsymbol{I}_D\right] X_\mathbf{s}^T \boldsymbol{y}\right)^{-\frac{2\alpha_\sigma + N}{2}} \tag{46}$$

Finally, we use the following steps to construct 'observed data' $(X, \boldsymbol{y})$ that we then plug into our posterior

- Let $x_1, \ldots x_5$ be 5 i.i.d random variables drawn from a uniform distribution over the discrete set $\{0, 1, 2\}$.

- Define the observed response $y_{\text{obs}} = \sum_{i=1}^{5} x_i$.

- 'Duplicate' the 5 covariates 3 times i.e. $x_j := x_{j \pmod 5)+1}$ for $j \in \{6, 7 \ldots, 20\}$.

- Repeat the above steps $N = 20$ times to obtain a 'design matrix' $X \in \mathbb{R}^{20 \times 20}$ and response $\boldsymbol{y} \in \mathbb{R}^{20}$. This design matrix and response can be obtained under the regression model in equation 41 by setting $\boldsymbol{\omega}$ to a vector of ones, and $\sigma = 0$ (i.e. noiseless regime).

The duplication of covariates induces multi-modality in the posterior $p(\mathbf{s} \mid X, \boldsymbol{y}_{\text{obs}})$, since masking out $x_1$ and leaving its copy $x_6$ unmasked is equivalent to masking $x_6$ and leaving $x_1$ unmasked.

## L  Persistent contrastive divergence (PCD)

The gold-standard approach for the estimation of statistical models is maximum-likelihood estimation (MLE). Unfortunately, for unnormalised models

$$\log p(\mathbf{s}; \; \boldsymbol{\theta}) = f(\mathbf{s}; \; \boldsymbol{\theta}) - \log Z(\boldsymbol{\theta}), \qquad\qquad Z(\boldsymbol{\theta}) = \sum_\mathbf{s} \exp(f(\mathbf{s}; \; \boldsymbol{\theta})) \tag{49}$$

the normaliser $Z(\boldsymbol{\theta})$ is presumed intractable, and so we cannot actually compute $\log p(\mathbf{s}; \; \boldsymbol{\theta})$, which is required for MLE. However, we can conveniently express the gradient as

$$\nabla_{\boldsymbol{\theta}} \log p(\mathbf{s}; \; \boldsymbol{\theta}) = \nabla_{\boldsymbol{\theta}} f_{\boldsymbol{\theta}}(\mathbf{s}) - \mathbb{E}_{p(\mathbf{s}; \; \boldsymbol{\theta})}[\nabla_{\boldsymbol{\theta}} f_{\boldsymbol{\theta}}(\mathbf{s})] \tag{50}$$

and then use a Monte-Carlo estimate of the second term, where approximate samples are drawn from $p(\mathbf{s}; \; \boldsymbol{\theta})$ using an MCMC sampler.

Running an MCMC sampler afresh every time we wish to perform a gradient-based parameter update quickly becomes prohibitively expensive. Persistent contrastive divergence provides a solution to this problem, by not starting afresh each time, but by *persisting* MCMC chains across parameter updates. One version of PCD is given in Algorithm L.6. This implementation adopts vanilla stochastic gradient descent to update the model parameters, but alternative optimisers (e.g. Adam (Kingma & Ba, 2014)) can be substituted.

---

**Algorithm L.6** Persistent contrastive divergence with buffer

---

**Require:** Discrete dataset $\mathcal{D}$. Integers $N_{\text{iters}}, N_{\text{batch}}, N_{\text{buffer}}$.

**Require:** Unnormalised log probability function $f(\,\cdot\,;\,\boldsymbol{\theta})$. Step-size $\epsilon$. Optional regulariser $h(\boldsymbol{\theta})$.

**Require:** MCMC transition operator $\mathcal{T}(\cdot \,|\, \cdot)$. Number of MCMC steps $K$.

$\quad \mathcal{B} \leftarrow [\mathbf{s}^1, \ldots \mathbf{s}^{N_{\text{buffer}}}]$           ▷ Initialise buffer of persistent chains

$\quad$ **for** $i \in \{1, \ldots, N_{\text{iters}}\}$ **do**

$\quad\quad$ Sample minibatch $B$ of size $N_{\text{batch}}$ from buffer $\mathcal{B}$

$\quad\quad$ **for** $j \in \{1, \ldots, K\}$ **do**

$\quad\quad\quad B \sim \mathcal{T}(\cdot \,|\, B)$           ▷ Parallelised update to minibatch

$\quad\quad$ **end for**

$\quad\quad$ Update buffer $\mathcal{B}$ with the new values in $B$           ▷ i.e. persist the chains

$\quad\quad$ Sample minibatch $X$ of size $N_{\text{batch}}$ from dataset $\mathcal{D}$

$$\boldsymbol{g} \;\leftarrow\; \frac{1}{N_{\text{batch}}} \left[ \sum_{\mathbf{s} \in X} \nabla_{\boldsymbol{\theta}} f(\mathbf{s};\,\boldsymbol{\theta}) \;-\; \sum_{\mathbf{s} \in B} \nabla_{\boldsymbol{\theta}} f(\mathbf{s};\,\boldsymbol{\theta}) \right] - \nabla_{\boldsymbol{\theta}} h(\boldsymbol{\theta}) \tag{47}$$

$$\boldsymbol{\theta} \;\leftarrow\; \boldsymbol{\theta} + \epsilon \boldsymbol{g} \tag{48}$$

$\quad$ **end for**

---

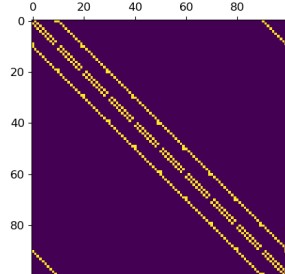 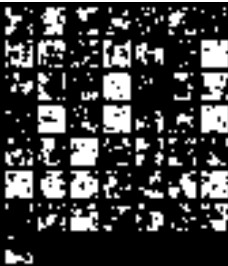

Figure 10: Left: ground truth lattice Ising model with connection strength $\theta = 0.2$. Right: samples from this model generated with $1,000,000$ steps of Gibbs sampling.

## M    Estimation of Ising models

The $100 \times 100$ lattice Ising matrix used in our experiments was generated via igraph, specifically, we call `igraph.Graph.Lattice(dim=[10, 10], circular=True)`[6], which returns a binary adjacency matrix, and then multiply this by a 'connection strength' parameter $\theta = 0.2$. The resulting matrix, along with samples from the Ising model, is shown in Figure 10.

For learning, we use PCD as described in Algorithm L.6, replacing vanilla SGD with Adam. The dataset $\mathcal{D}$ consists of $10,000$ samples generated via $1,000,000$ steps of Gibbs sampling. We set $N_{\text{iters}} = 2,000, N_{\text{batch}} = 50, N_{\text{buffer}} = 5000, \epsilon = 0.0003$. Our model takes the form $f(\mathbf{s};\, J) = \mathbf{s}^T J \mathbf{s}$, where $J$ is a symmetric matrix. We use the regulariser $h(J) = 0.01 \sum_{i,j} |J_{i,j}|$.

---

[6]We use version `0.9.8` of the `igraph` package.

## N Sampling Lattice Ising models with controllable higher order interactions

The motivation for PAVG was to incorporate second-order interactions into the sampling process. We expect this to be of clear benefit when the target distribution is log-quadratic, and that is indeed the case, as shown in the Ising model experiments of Section 5.3. It is interesting to consider how the performance of PAVG changes as we gradually add higher-order terms to such an Ising model. As a simple experiment, we consider sampling from a low-dimensional Lattice Ising target distribution ($d = 16$) defined by

$$f(\mathbf{s}) = \mathbf{s}^T J \mathbf{s} + \alpha \sum_{(i,j,k) \in \mathcal{I}} \mathbf{s}_i \mathbf{s}_j \mathbf{s}_k, \tag{51}$$

where $\mathcal{I}$ is an index set containing 50 randomly selected triples and the matrix $J$ is as described in Appendix M. We use a low-dimensional model so that, similar to our Bayesian regression experiment, we can analytically compute all pairwise marginals and use the estimation error of such marginals (as defined in the bottom right quadrant of table 5) for evaluation.

We compare PAVG (with preconditioning matrix $J$) with AVG for different settings of $\alpha$. AVG is a natural comparison method, as it is a special case of PAVG that does not have access to the preconditioning matrix. The results are shown in Figure 11.

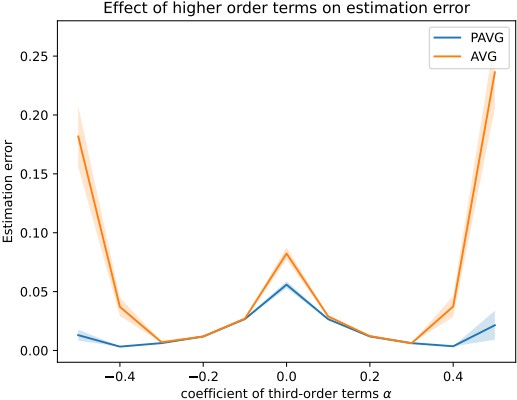

Figure 11: Estimation error for different values of $\alpha$ when sampling from a target distribution given by equation 51.

When the coefficient $\alpha$ is 0, we simply have an Ising model, and PAVG outperforms AVG as expected. For small $\alpha$ (in absolute value), PAVG and AVG perform similarly, whilst for larger values of $\alpha$, PAVG again outperforms AVG. This U-shaped performance gap is somewhat surprising, and we do not claim to have a full explanation of why it occurs. However, we believe it might be understood in terms of the relative speed of PAVG versus AVG during the two phases of MCMC sampling: the *burn-in* period and, afterwards, the *exploration of the typical set*.

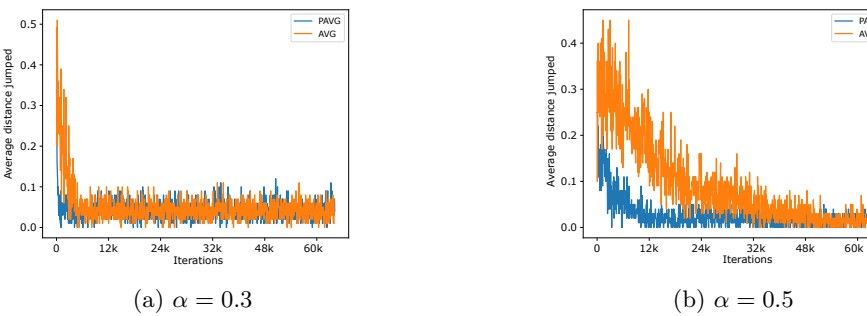

(a) $\alpha = 0.3$        (b) $\alpha = 0.5$

Figure 12: Average L1 distance moved (across 100 parallel chains) for two different values of $\alpha$.

Our conjecture is that, in the specific context of Ising models with non-negligible third-order terms, the preconditioning in PAVG isn't very useful for faster exploration of the typical set, but it is useful during the burn-in period. Some evidence for this claim can be found in Figure 12, where we plot the average 'jump distance' ($\|\mathbf{s}_t - \mathbf{s}_{t+1}\|_1$) between successive states in the MCMC chains for $\alpha = 0.3$ and $\alpha = 0.5$. We see that, for both values of $\alpha$, PAVG and AVG eventually converge to a steady-state where the jump distance is low, which may indicate that the typical set has been reached and is now being explored. In both cases, PAVG is faster to reach this steady-state suggesting that it has a smaller burn-in period. However, PAVG is only *marginally* faster than AVG when $\alpha = 0.3$, perhaps because the initial distribution of MCMC chains is already reasonably close to the target distribution, so neither method has difficulty. In contrast, PAVG is substantially faster when $\alpha = 0.5$.

One corollary of our conjecture is that, for $\alpha = 0.5$, the estimation error for AVG ought to *eventually* come down if we run it for longer. Indeed, running it for twice as long (120k iterations) does reduce the error to the much lower value of 0.08 (although this is still more than twice the error of PAVG). We believe that further analysis of these results is beyond the scope of this appendix, and leave it to future work. Regardless of whether our above conjecture holds, a key take-away message from Figure 11 is that preconditioning with second-order interactions can be highly beneficial in the presence of strong higher-order interactions. Indeed our other experiments on sparse Bayesian regression and deep energy-based models also validate this claim.

## O    Sampling convolutional energy-based models

The architecture of the convolutional neural network used in our experiments is shown in Table 14a, alongside images from the USPS dataset and our ground-truth EBM. To learn the ground-truth EBM model $\frac{1}{2}\mathbf{s}^T J^* \mathbf{s} + f^*(\mathbf{s})$, we use PCD as shown in Algorithm L.6 with GWG as our sampler and $K = 50$. We set $N_{\text{iters}} = 10,000, N_{\text{batch}} = 50, N_{\text{buffer}} = 5,000, \epsilon = 0.0003$. We use weight decay of 0.0001 on the neural net weights.

After learning this ground-truth EBM, we rerun PCD using the model $\frac{1}{2}\mathbf{s}^T J \mathbf{s} + f^*(\mathbf{s})$, where $J$ is a symmetric matrix of parameters. During this re-estimation phase, the settings of PCD remain the same, except for $N_{\text{iters}} = 2,000$.

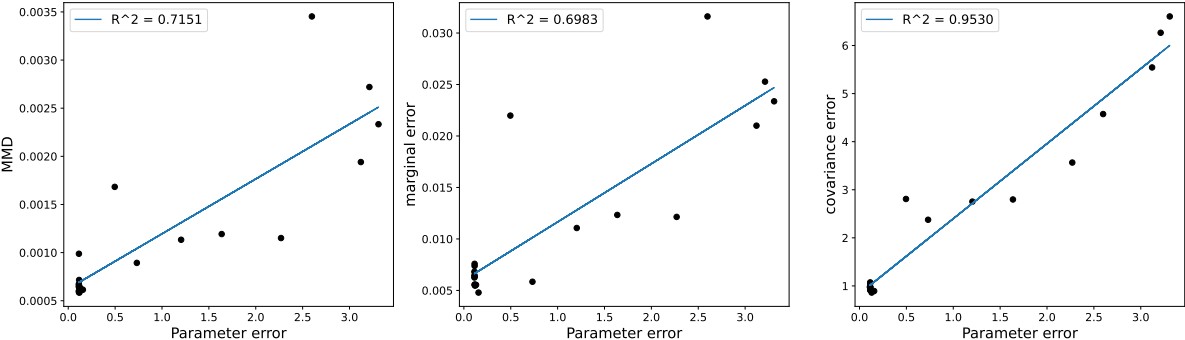

Figure 13: Comparison of the parameter estimation error metric $\|J - J^*\|_F$ (as reported in Table 2) with 3 different sample-based metrics. Each point in a figure corresponds to a cell in Table 2 i.e. it corresponds to a particular choice of MCMC method and value of $K$ (thus, there are 20 points per subplot). These sample-based metrics all compare the similarity of two sets of samples. Here, we compare the 5K buffer samples produced by a particular MCMC method against 5k samples from the ground-truth EBM. Maximum mean discrepancy (MMD) (Gretton et al., 2012) is computed using the same code as Grathwohl et al. (2021). 'marginal error' is the absolute difference in empirical means (averaged over all dimensions). 'covariance error' is the Frobenius norm of the difference between the empirical covariance matrices computed using each set of samples.

| |
|---|
| Conv2d(1, 16, 3, 1, 1) |
| SiLU() |
| Conv2d(16, 32, 4, 2, 1) |
| SiLU() |
| Conv2d(32, 32, 3, 1, 1) |
| SiLU() |
| Conv2d(32, 64, 4, 2, 1) |
| SiLU() |
| Conv2d(64, 64, 3, 1, 1) |
| SiLU() |
| Conv2d(64, 128, 4, 2, 1) |
| SiLU() |
| Conv2d(128, 128, 2, 1, 0) |
| SiLU() |

(a) 7-layer convolutional neural network used as an EBM to model USPS digits. The SiLU() activation function (Ramachandran et al., 2017) is also called the 'swish' activation.

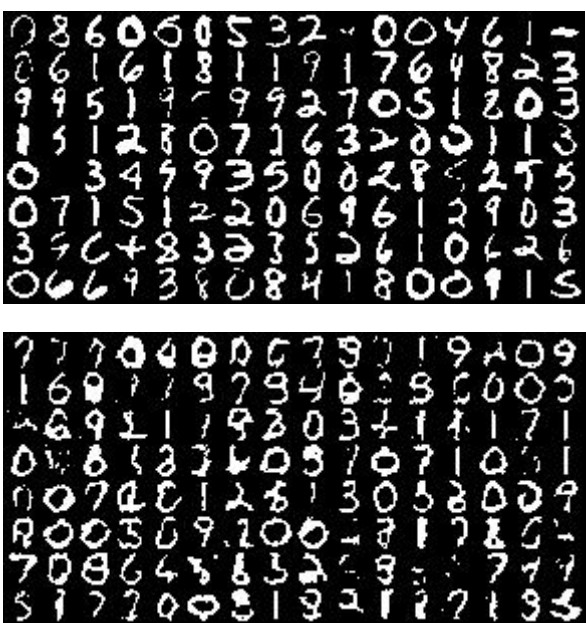

(b) Top row: real images from the USPS dataset. Bottom row: samples from ground-truth quadratic-EBM model trained with GWG and $K = 50$.

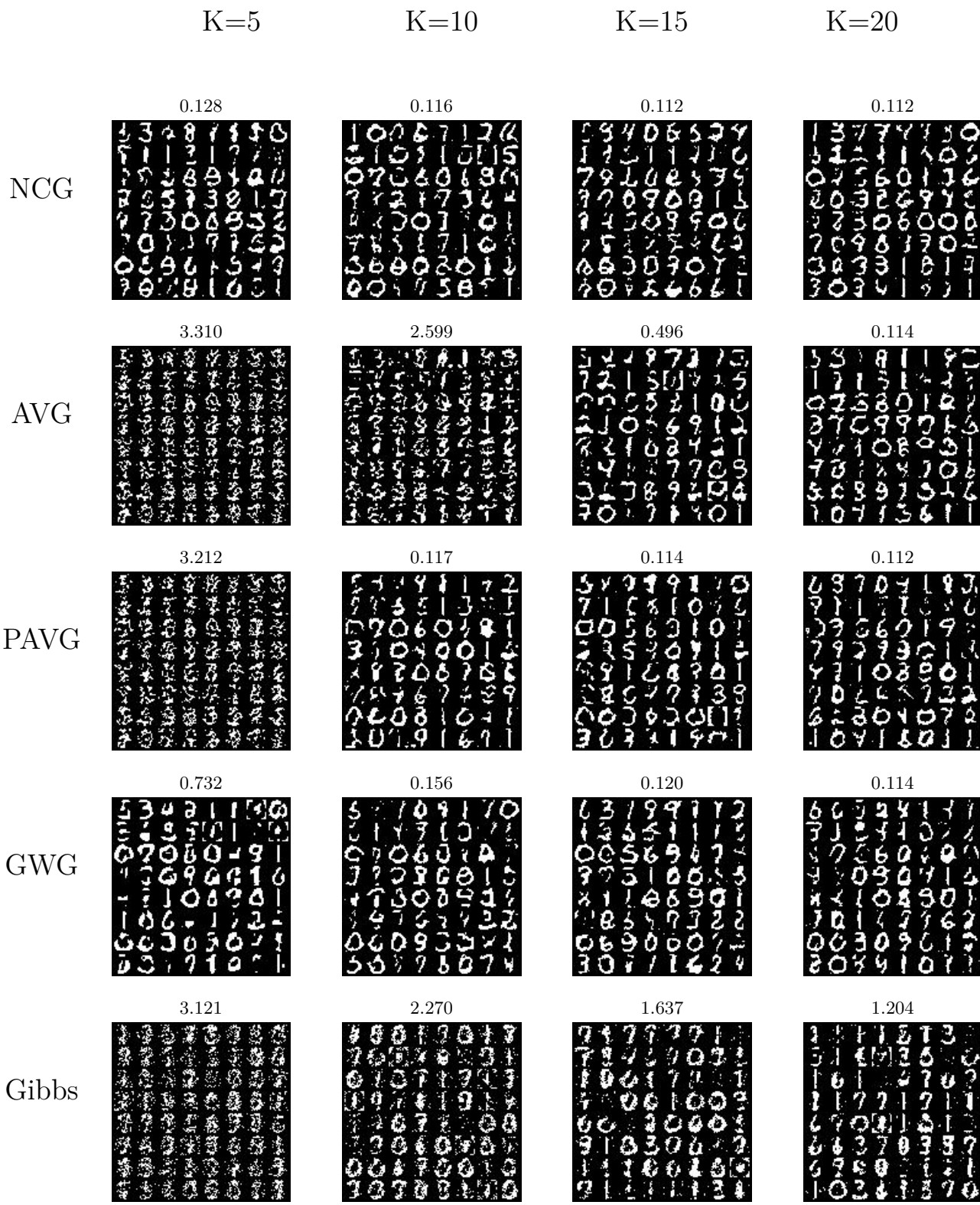

Figure 15: Samples for the methods described in Table 2. Above each image is the corresponding estimation error.

