# OpenReview forum: "Enhanced gradient-based MCMC in discrete spaces"
_TMLR — Accepted by TMLR_

### Review · Reviewer_UqhQ · 2022-08-29

**Summary Of Contributions:**

The paper proposes gradient-based MCMC methods to discrete problems. The paper is based on the idea of interpreting discrete spaces as restrictions of an underlying continuous field, and utilising their gradients. The paper then extends such gibbs-with-gradient MCMC method by applying similar treatment to MALA and its preconditioned version.


**Broader Impact Concerns:**

No issues

**Requested Changes:**


* Please specify if “f” (or s) is known, unknown, or possible to evaluate. Define what it means that “f is restriction of a differentiable function [which one?]”. Define what it means to take a gradient of a "restriction". The background gives the impression that the paper is written as a companion to Gratwohl’21, while it should be self-contained. Similarly in eq 6 the discrete f changes into continuous f, and gradient is taken, but with no explanation what this means mathematically. The paper needs to give a precise and formal presentation of f and nabla f.
* The discussion of cost of eq 6 is unclaer, and I’m not sure what is costly and what is cheap about eq 6. I assume that entire eq 6 is cheap to evaluate, but doing search around the q(s|..) is expensive. Please elaborate.
* Eq 6 is in multivariate form where it seems to directly support multivariate updates to s_t. Yet, the text claims that one can only update one dimension at a time. Please explain.
* The sec 2.4. last phrase implies that linearising the target AND using symmetric kernels can’t account for non-linear target interactions. Why do we need the “and” here? Surely linearising something alone removes non-linear effects? Does the symmetric kernels also pose problems?
* “as discussed, use of hamming balls in gwg is challenging..” I don’t recognize this discussion. I assume it refers to post-eq-6 paragraph, but it does not explain why increasing the radius is a problem. If you have a linear approximation around center point s_t, surely one can continue the linear plane for as far away as one wants with little expense. Surely one can also make it multidimensional for cheap as well? Please make it explicit where is the bottleneck here. Furthermore, I really don’t see how changing a hamming ball to a gaussian ball changes the cost of anything: surely both are equally tractable and fast to compute.
* “if we restrict s to discrete space” (before eq8) implies that eq 7 did not restrict f(s) to discrete space. This makes me very confused: did we have a fully continuous f(s) in eq 7? Is this a typo? Please explain what is the nature of f and s.
* I have hard time understanding eq 8. I don’t understand how we go from 7 to 8. I don’t get why restriction leads to taking a product over dimensions. In eq 7 it was a sum.. I assume that we are completing the square here, but it seems that the s_t^2 term is missing. Please make this understandable.
* It seems that eq 8 is a MALA-version of eq 6. The main bottleneck was that you couldn’t broaden the hamming ball. I don’t see what eq8 improves. Please explain why the eq8 solves the problem(s) of eq6.
* 3.2.: I don’t think f(s) has been defined for s \in R. Eq1 only defines f(s) for discrete s, so thus the pi(s,z) is undefined. The fluid nature of {s,p,f} as continuous or discrete things is very confusing. I would suggest separating the disc/cont versions notationally, eg. by using \tilde.
* Can you motivate or explain the N(z | s/sqrt,I) term? I can’t follow why are we dividing the mean with square root. It would also be good to explain what z is or tries to model or conceptualise. It seems that z is some kind of scaled version of s, and s is then following z’s. Perhaps this introduces some lag or momentum to the system? Not sure.. Please explain.
* In eq 9 the taylor expansion over f(s) leads to the eq 9, but previously also eq 6 was a taylor expansion of f(s) as well, and it lead to a different form without the lone f(s_t) term. Are these different kinds of Taylors? Please explain.
* eq 11 is missing \cal{N}
* eq 13: this looks like second order taylor, but is not. Can we make this kind of approximation? Surely this only applies if M is carefully chosen. However, there is no discussion of what M is. Please elaborate. Later it is claimed that we can use empirical covariance or precision as M. I don’t understand this. M should be a hessian matrix or second derivatives. Surely covariates have little to do with higher order gradients. One should also explain what happens if the eq13 is not satisfied: isn’t this a problem?
* fig 1 right seems odd. Why do we increase the sampling probably towards corners? Is this intentional? Does the sampling probably keep increasing as a function of distance? The fig1 right needs to zoom out and also show regions where the probably is decreasing or vanishing.


**Strengths And Weaknesses:**

The paper is well written, generally easy to follow, and shows expertise from the authors. The two new samplers are generally well derived and presented, and the experiments clearly demonstrate the benefits of the new methods in consistent way. The paper suffers from some (minor) clarity issues, listed below. After these have been adressed, I'm happy to see this accepted.

---

> ### Author Response · Authors · 2022-09-14
> **Response (part 1)**
>
> Thank your detailed comments and suggestions. We have made many revisions to the main text (see revised pdf) based on them. We list them here one-by-one:
>
> > Please specify if “f” (or s) is known, unknown ...
>
> Thank you for raising this. We have updated Section 2 to address your points. The reviewer is correct that are arbitrarily many continuous extensions of the discrete function f, and so the question of `which one?' needs answering. We have hopefully clarified this in the revised pdf by stating that the continuous and discrete functions have the exact same mathematical expression; it is only the domain that changes. We believe our paper now has the same level of specificity and formality as Grathwohl 21.
>
> > The discussion of cost of eq 6 is unclear, and ...
>
> We now discuss this in Section 2.4 of the revised pdf and in further detail in the new Appendix C.
>
> > The sec 2.4. last phrase implies that linearising the target AND using symmetric kernels ...
>
> You are correct that the emphasis on *symmetric* kernels was wrong, and we have ammended the text. However, as we discuss in Section 3.1.1., it is possible to account for second-order interactions in the target distribution through particular choices of kernel (whilst still using a first-order Taylor expansion of f(s)). Indeed, PMALA does exactly this. Hence, the choice of kernel does matter, and we still need the "and" in sec 2.4.
>
> > “as discussed, use of hamming balls in gwg is challenging..” I don’t recognize this discussion. I assume it refers ...
>
> We agree that our original exposition was lacking enough detail. We have added substantially more details in a new Appendix C ("The computational benefit of NCG over GWG with large Hamming balls"), which we reference in Section 2.4.
>
> > “if we restrict s to discrete space” (before eq8) implies that eq 7 did not
>
> We agree that this statement was confusing and have ammended it. The new wording hopefully conveys that it is a feature (and not a bug) of Eq 7 that its domain is unspecified. We want to say that *if* the domain is continuous, we obtain MALA, and *if* it is discrete, we obtain NCG.
>
> > I have hard time understanding eq 8. I don’t understand how ...
>
> Indeed, the factorisation has nothing to do with a restriction (our previous wording was unclear, apologies). We have added a footnote which explains that, when rearranging eq 7, we can effectively drop terms that do not depend on $s$ (such as the $s_t^2$ term) because they act as multiplicative constants on the unnormalised distribution, and therefore get dealt with when we normalise it. If this explanation hasn't cleared things up, then please let us know.
>
> > It seems that eq 8 is a MALA-version of eq 6. The main bottleneck
>
> We have reworded Section 3.1 to better explain the benefits of eq8, and added additional explanation of the benefits in the new Appendix C.
>
> > in Section 3.2. I don’t think f(s) has been defined for $s \in R$ ...
>
> We have altered the beginning of Section 2 to ensure that both p and f are well-defined for $s \in R^d$. Moreover, we have reworded parts of Section 3.2 to more clearly signpost what domain we are operating in. However, we believe that the re-use of the notation is crucial to our exposition. One of our key messages is that we can import ideas from continuous spaces into discrete spaces by re-using the same functional-forms/mathematical expressions, and only altering the domains over which they are defined. Re-use of notation helps us to convey this message.
>
> > Can you motivate or explain the $\mathcal{N}(z | s / \sqrt{...}, I)$ term? ...
>
> Thank you for this suggestion. We have added a new appendix D ("Interpretation of auxiliary variables") that tries to provide some clarity on this matter.
>
> > In eq 9 the taylor expansion over $f(s)$ leads to the eq 9, but previously ...
>
> In unnormalised conditional distributions of $s$ given $s_t$, we can always drop terms that do not depend on $s$ (such as the $f(s_t)$ term) because they act as multiplicative constants on the unnormalised distribution. In Equation 6 we follow Grathwohl '21 by not including $f(s_t)$ and in equation 9 we follow Titsias et al. '18 by including it.
>
> > eq 11 is missing $\cal{N}$
>
> Thanks for this correction.

---

> ### Author Response · Authors · 2022-09-14
> **Response (part 2)**
>
> > eq 13: this looks like second order taylor, but is not. Can we make this kind of approximation? ...
>
> We disagree that "there is no discussion of what M is"; we discuss it in section 3.3.1 named "Choice of preconditioning matrix M". We do agree that this is an unusual kind of approximation, and that one might expect us to use a second-order Taylor approximation. We have added more details to Section 3.3.1 explaining why we deviate from a second-order Taylor. We agree that there are no strong apriori/theoretical reasons for expecting our approximation (with an adaptively scaled covariance/precision matrix) to be especially useful. But empirically, it does yield substantial benefits compared to AVG, which is arguably the most natural baseline since AVG is a special-case of PAVG in which the M matrix is zero and hence we revert back to a first-order Taylor approximation. Moreover, as we say in the text our adaptive re-scaling means "we can automatically ‘fallback’ to AVG by learning $\gamma = 0$". In practice however, we found that this `safety net' did not get activated.
>
> > fig 1 right seems odd. Why do we increase
>
> Yes, this is intentional. The figure depicts the *entire* discrete state-space that we are trying to sample from; zooming out would not reveal any more of the state-space. When sampling from continuous distributions in $R^d$, it is correct that probability density must decrease and vanish as one 'zooms out'. In discrete spaces, this is not true. For instance, suppose we have a model of the form $f(s) = s^T J s$, where J is positive definite. If we were to plot this function for real-valued s, then it would keep increasing as we `zoom out' and its integral (over $R^d$) would be infinite. However, for binary-valued s, it defines a perfectly valid unnormalised model.
>
> We hope we have adequately addressed your questions and concerns, and thank your for your detailed comments.

---

### Review · Reviewer_MRuB · 2022-09-01

**Summary Of Contributions:**

The authors consider the problem of sampling in combinatorial spaces using gradient-based methods that have seen a recent surge in interest due to works of Zanella (2020) on locally-balanced proposals and Grathwohl et.al (2021) on Gibbs with Gradients (GwG).

The authors identify two main problems with GwG, namely, inability to update multiple dimensions concurrently and reliance on only first order structure of the problem ignoring higher-order interactions. The main contributions of the paper are to address these shortcomings.

The authors achieve this by first extending the ideas of Grathwohl et.al (2021) to other continuous gradient based samplers like MALA and Preconditioned-MALA to propose Norm-constrained Gradient sampler which can update multiple dimensions at once and Auxiliary Variable Gradient + Preconditioned AVG samplers based on the auxiliary sampler proposed by (Titsias & Papaspiliopoulos) which is capable of incorporating second-order information. They show that the proposed samplers have superior performance to GwG on simple experiments.

**Requested Changes:**

Please see my comments above for the details of my points below.
- Adding relevant citations
- adding details in section 3.1.1
- discuss computational complexity of the methods
- (good to have but I won't consider it necessary at this stage) additional experiment on a real-world application or more complex dataset

**Strengths And Weaknesses:**

**Strengths:**
- The ideas to extend GwG to MALA type samplers to overcome the drawbacks of GwG are interesting and clever. I also appreciated the authors discussion on challenges faced by these discrete samplers when incorporating preconditioning thereby showing that it is non-trivial to achieve in the discrete space.
- The algorithms have been tested on a wide variety of tasks (albeit simpler and more synthetic in nature) demonstrating its performance on a diverse set of problems.
- The paper is also well written and easy to follow. Sufficient explanation have been provided at appropriate places to make the arguments clear. I'd also commend the authors on providing intuitive explanations for most ideas (I wish more ML papers did this).

**Weakness:**
- I think a weakness of the paper is the restriction of experiments on relatively easier tasks on either synthetic or simpler datasets. As I mentioned earlier, the authors did a good job to show the performance of the algorithms on diverse tasks. I also understand that these tasks have become commonplace in the literature. However, I believe the paper would have benefitted greatly from considering at-least one task closer to a real-world application or based on a more complicated dataset. For example, Zhang et.al 2022 who concurrently proposed the NCG sampler considered the task of text infilling. Currently, I find the experiments to be limited to relatively lower dimensions and simpler tasks to be interesting beyond as just proofs-of-concept.
- A motivation for authors with P-AVG was to incorporate higher-order interaction terms. While, the experiments and the performance of P-AVG *can imply* that incorporating such information is useful to the performance of the sampler, I wonder if it might have been useful to construct a more controlled experiment to develop this point. For example, the authors can extend their Ising model experiment to create a frustrated system on a square lattice where the nodes can have both long-range interactions as well as second-order couplings. Sampling from such a system can be beneficial to more directly isolate the benefit of incorporating second-order interactions.
- *Eq.8*: the normalization term here still seems expensive to me to compute or am I missing something here? How well will it scale to increasing dimensions? In any case, I think it might be useful to explicitly give the computational costs for the methods to keep things transparent and for future works in the community.

*minor points and comments*:
- It would be easier to understand the gains made the proposed samplers if the code was available through some anonymous link.
- Through pre-conditioning the authors were able to incorporate second-order interactions. I am curious if the authors have any intuition to extend this to even higher order interactions which might be useful for many problems in Physics like in spin-glass systems? Specifically, in problems which has such higher-order interactions, do the authors have a sense of how well the proposed method will work or how badly they will fail?
- In Section 3.1.1, the authors provide a discussion on why preconditioning is difficult for NCG. I think the subsection may benefit from a more detailed exposition on how the preconditioning lead to a pairwise MRF and show its intractability. This will help make the manuscript more accessible.
- In section 4, the authors discuss methods where discrete space is embedded in a continuous one and highlight their drawbacks. The authors miss the ref "Sampling in Combinatorial Spaces with SurVAE Flow Augmented MCMC, AISTATS 2021" which addresses some of the drawbacks mentioned in the section.
- Another small question: in the algorithms box you mention the step of sampling from the proposal distributions you have created. However, it is not immediately clear to me how you sample from say Eq.8? Is it similar to what is done in GwG? If not, can you please add more details around this

---

> ### Author Response · Authors · 2022-09-14
> **Response**
>
> Thank you for your comments and suggestions. We address them one by one:
>
> > I think a weakness of the paper is the restriction of experiments on relatively easier tasks on either synthetic or simpler datasets ... Currently, I find the experiments to be limited to relatively lower dimensions and simpler tasks to be interesting beyond as just proofs-of-concept.
>
> We appreciate that this would strengthen the paper, but do not think it is feasible in the short time-window available to us. We did run a new simulation following your other suggestion (please see below).
>
> > A motivation for authors with P-AVG was to incorporate higher-order interaction terms ... I wonder if it might have been useful to construct a more controlled experiment to develop this point ...
>
> Thank you for this suggestion. We have run a new experiment based on your idea of an Ising model with higher-order interactions as explained in Appendix N. We would like to emphasise that the results are preliminary, and that the analysis is therefore somewhat conjectural. Nevertheless, we believe it sheds some light on the benefits of PAVG.
>
> > Eq.8: the normalization term here still seems expensive to me to compute or am I missing something here? How well will it scale to increasing dimensions? ... Another small question: in the algorithms box you mention the step of sampling from the proposal distributions you have created. However, it is not immediately clear to me how you sample from say Eq.8? Is it similar to what is done in GwG?
>
> We have added a new appendix C ("The computational benefit of NCG over GWG with large Hamming balls"), which hopefully answers both of these questions. In that appendix, we explain how normalisation and sampling is performed for GWG versus NCG, and the associated costs of these operations.
>
> > It would be easier to understand the gains ... if code was available ...
>
> Unfortunately, we did not find the time to make the code suitable for public release in the short period available. We hope that the revisions to the main text, in combination with the new appendix C, make the gains of the proposed samplers more obvious. We will of course make the code available upon publication.
>
> > Through pre-conditioning the authors were able to incorporate second-order interactions. I am curious if the authors have any intuition to extend this to even higher order interactions ...
>
> We are not familiar with the assumed physics background implicit in this question, so it difficult for us to speculate on whether our methods could be extended to handle such higher-order interactions. We can say that many of our experiments (4th-order ordinal, sparse Bayesian regression, convolutional EBM) involve higher-order terms, and thus the experimental results we have provided do shed light on the question of `how well the proposed method will work or how badly they will fail?' for challenging, non-quadratic target distributions.
>
> > In Section 3.1.1, the authors provide a discussion on why preconditioning is difficult for NCG. I think the subsection may benefit from a more detailed exposition on how the preconditioning lead to a pairwise MRF and show its intractability.
>
> We have slightly expanded our explanation in 3.1.1, which we hope addresses your concerns. Please let us know if this new explanation is missing something.
>
> > The authors miss the ref "Sampling in Combinatorial Spaces with SurVAE Flow Augmented MCMC, AISTATS 2021" which addresses some of the drawbacks mentioned in the section.
>
> Thank you for raising this to our attention, we have added this to our related work section.
>
> Overall, we hope we have adequately addressed your requested changes for the paper, and thank you for your constructive feedback.

---

### Review · Reviewer_QtjV · 2022-09-02

**Summary Of Contributions:**

The paper presents three methods for sampling from intractable distributions in discrete
spaces. The problem is well-motivated: such situations are ubiquitous and sampling from
discrete distributions is significantly more difficult than sampling from Euclidean
spaces. The authors develop principled MCMC-based methods adopted from similar methods
developed in prior work in different settings.

While I am not an expert on this area, I am cautiously optimistic about the paper. I
believe the paper can be accepted after some suggested revisions below.

**Requested Changes:**

See aboveNo

**Strengths And Weaknesses:**

Novelty: The methods introduced in the paper are appear to be a straightforward
combination and/or extension of existing ideas. As TMLR prioritizes correctness over
novelty, this is not a significant issue in and of itself.

Experiments: The comparison with other methods are thorough, although I would have liked
to see the recent method of Zhang et al '22. Any reason why this was not included in the
evaluation?

My main criticism of the paper in its current form is the lack of sensible defaults for
the hyperparameters and an analysis of sensitivity to hyperparameters for the proposed
methods and the baselines. I believe this is important to justify the authors' claims tha
t
their method performs better than prior methods on the problem.
In my experience using probabilistic programming tools, a fair amount of tuning is
required to get MCMC methods working efficiently. The authors have specified the specific
hyperparameter choices in Table 3, but have not commented on how much effort was spent to
tune each method for each problem. Ideally, the authors would show results for all method
s
on several hyperparameter choices.


Other questions:

Equation 8: It appears sampling from q requires a summation over all points in the
discrete space. This can be computationally prohibitive in high dimensional settings.
Is this avoidable in an implementation? If not, how come your method does well in
practice?

---

> ### Author Response · Authors · 2022-09-15
> **Response**
>
> Thank you for your feedback. We have revised the pdf in response to all reviewers' feedback, highlighting the new material in red. To respond to your questions:
>
> > Experiments: The comparison with other methods are thorough, although I would have liked to see the recent method of Zhang et al '22. Any reason why this was not included in the evaluation?
>
> As explained in Section 4, the *concurrent* work of Zhang et al '22 introduces a sampler (DMALA) that coincides exactly with one of our samplers (NCG). Thus, we have already included their method in our experiments.
>
> > My main criticism of the paper in its current form is the lack of sensible defaults for the hyperparameters and an analysis of sensitivity to hyperparameters for the proposed methods ...
>
> Thank you for raising this point; we agree that analysing the sensitivity of step-size parameters is important and have included additional simulations in the expanded Appendix H (subsection H.1). These simulations were performed for all methods (NCG, AVG and PAVG) over a range of step-sizes spanning two orders of magnitude. We believe the results demonstrate a decent amount of robustness in our methods; in particular, NCG and PAVG show improvement over GWG for a wide range of step-sizes. We do not fully agree that we "have not commented on how much effort was spent to tune each method for each problem", since Appendix H did outline the search procedure used to select the step-sizes. However, we have added more detail to that procedure, which hopefully makes it clearer.
>
> > Equation 8: It appears sampling from q requires a summation over all points in the discrete space. This can be computationally prohibitive in high dimensional settings. Is this avoidable in an implementation? ...
>
> The distribution in Equation 8 factorises across all dimensions, meaning we can separately normalise and sample $d$ *one-dimensional* distributions in parallel. Indeed, the only summation that appears in Equation 8 is over $\mathcal{S}$, which is a 1-dimensional set of points e.g. {0, 1} if we are working with binary random variables. We have added an explanation of this in a new appendix C ("The computational benefit of NCG over GWG with large Hamming balls"). In that appendix, we also explain how normalisation and sampling is costly for GWG when using large Hamming balls.
>
> Again, we thank you for your questions/suggestions, and hope that we have adequately addressed them.

---

### Author Response · Authors · 2022-12-07
**Code is available**

https://github.com/benrhodes26/enhanced_discrete_gradient_mcmc

---

### Decision · Action_Editors · 2022-10-04

**Recommendation:** Accept as is

**Comment:**

All three reviewers were satisfied with the efforts and responses put forward by the authors during the response phase, which cleared up some minor outstanding issues raised by the reviewers during the initial review phase. There is consensus that this is a strong contribution and universal support for the publication of the revised manuscript.

**Audience:**

Yes, there is no question that the material in this manuscript would be of interest to at least a portion of the TMLR audience. MCMC is a foundational tool for a large segment of the readership.

**Claims And Evidence:**

Yes. The claims made in the manuscript are supported by clear and detailed exposition, with strong motivation and solid arguments made for each component of the proposed methods. The performance of these methods is further supported empirically through a series of detailed and well-designed experiments.